# Super-resolution imaging of a 2.5 kb non-repetitive DNA *in situ* in the nuclear genome using molecular beacon probes

Yanxiang Ni[1,2,3]*[†], Bo Cao[1†], Tszshan Ma[2†], Gang Niu[1,4,5†], Yingdong Huo[1], Jiandong Huang[3], Danni Chen[1], Yi Liu[1], Bin Yu[1], Michael Q Zhang[2,6]*, Hanben Niu[1]*[‡]

[1]Key Laboratory of Optoelectronic Devices and Systems of Ministry of Education and Guangdong Province, College of Optoelectronic Engineering, Shenzhen University, Shenzhen, China; [2]MOE Key laboratory of Bioinformatics; Bioinformatics Division and Center for Synthetic and Systems Biology, TNLIST; School of Medicine, Tsinghua University, Beijing, China; [3]School of Biomedical Sciences, The University of Hong Kong, Hong Kong, China; [4]LemonData Biotech, Shenzhen, China; [5]Phil Rivers Technology, Beijing, China; [6]Department of Biological Sciences, Center for Systems Biology, The University of Texas at Dallas, Dallas, United States

*For correspondence:
niyanxiang0000@gmail.com (YN);
michaelzhang@tsinghua.edu.cn
(MQZ); hbniu@szu.edu.cn (HN)

[†]These authors contributed equally to this work

[‡]Deceased

Competing interests: The authors declare that no competing interests exist.

**Abstract** High-resolution visualization of short non-repetitive DNA *in situ* in the nuclear genome is essential for studying looping interactions and chromatin organization in single cells. Recent advances in fluorescence *in situ* hybridization (FISH) using Oligopaint probes have enabled super-resolution imaging of genomic domains with a resolution limit of 4.9 kb. To target shorter elements, we developed a simple FISH method that uses molecular beacon (MB) probes to facilitate the probe-target binding, while minimizing non-specific fluorescence. We used three-dimensional stochastic optical reconstruction microscopy (3D-STORM) with optimized imaging conditions to efficiently distinguish sparsely distributed Alexa-647 from background cellular autofluorescence. Utilizing 3D-STORM and only 29–34 individual MB probes, we observed 3D fine-scale nanostructures of 2.5 kb integrated or endogenous unique DNA *in situ* in human or mouse genome, respectively. We demonstrated our MB-based FISH method was capable of visualizing the so far shortest non-repetitive genomic sequence in 3D at super-resolution.

## Introduction

Being able to visualize short unique genomic sequences *in situ* at high resolution is essential to understanding looping interactions and chromatin organization in single cells, but chromosome conformation capture (3C)-based methods can only reveal three-dimensional (3D) chromatin interactions as the cell population average (*Guo et al., 2015*; *Tang et al., 2015*; *Dixon et al., 2012*; *Smith et al., 2016*; *Dekker and Misteli, 2015*). Various techniques have been developed to address this issue, such as oligonucleotide-based fluorescence *in situ* hybridization (FISH) and genomic locus targeting with optimized CRISPR/Cas system (*Hogan et al., 2015*; *Chen et al., 2013*; *Beliveau et al., 2012*; *Anton et al., 2014*; *Boyle et al., 2011*; *Yamada et al., 2011*). However, such methods are mainly constrained by their limited ability to target the short non-repetitive element of interest or to image short unique DNA at high-resolution.

A recent study combined oligopaint probe-based FISH (Oligopaint-FISH) with stochastic optical reconstruction microscopy (STORM) to enable super-resolution visualization of genomic domains at a sequence resolution limit of 4.9 kb (*Boettiger et al., 2016*; *Beliveau et al., 2015*). STORM is a

high-resolution single-molecule imaging technique, which relies on the stochastic activation and localization of many individual photo-switchable fluorescent dyes (*Bates et al., 2005*). Targeting a genomic region requires hundreds to thousands of Oligopaints that are derived using a complicated probe generation system. Furthermore, high-resolution imaging of Oligopaint-labeled targets requires a secondary probe and the pairing of fluorophores on both probes to achieve sufficient numbers of localizations needed to resolve fine-scale nanostructures (*Thompson et al., 2002*; *Huang et al., 2010*; *Deschout et al., 2014*; *Beliveau et al., 2015*). Short genomic elements have been shown to play many essential roles in genomic functions (*Blinka et al., 2016*; *Hogan et al., 2015*), but high-resolution visualization of elements smaller than the resolution limit of the above method is currently not possible. For visualizing shorter elements, we developed a simple super-resolution FISH method that uses molecular beacon (MB) probes (MB-FISH) to allow specific binding between probes and target sequences at a relatively low hybridization temperature, while reducing non-specific binding. We optimized the imaging conditions of the 3D-STORM to efficiently recognize sparsely distributed Alexa-647 from the background cellular autofluorescence. Utilizing 3D-STORM and only 29–34 different MB probes, we were able to observe fine-scale 3D structures of 2.5 kb integrated or endogenous non-repetitive DNA *in situ* in the nuclear genome of human or mouse, respectively. We demonstrated our MB-FISH was capable of resolving the as yet shortest unique genomic sequence in 3D at the nanoscale resolution.

## Results

### MB probe design minimizes non-specific fluorescence and facilitates probe-target binding

Because the mammalian genome is long and complex, labeling a non-repetitive genomic sequence using FISH requires minimizing off-target binding. One common solution is to increase the FISH hybridization temperature (*Beliveau et al., 2015*, *2012*), but this leads to instability in the probe binding. To decrease non-specific fluorescence, we designed FISH probes in the form of molecular beacons that carry a fluorescent dye and quencher at separate ends. When not bound to the target sequence, these probes assume a hairpin configuration that turns off the fluorescence (*Tyagi and Kramer, 1996*). Alexa-647 was chosen as the fluorescent dye due to its well-documented ability to switch between dark and fluorescent states when used in STORM imaging (*Dempsey et al., 2011*; *Heilemann et al., 2009*; *Olivier et al., 2013*). Black hole quencher 3 (BHQ3, a specific quencher for Alexa-647) (*Didenko, 2001*), was tethered to the other end of probe to quench the fluorescence of the MB when unbound or non-specifically associated, which ensures the MB probe fluoresces only upon binding with its target (*Figure 1a*, top). Using a specifically designed set of MB probes, non-repetitive DNA can then be fluorescently distinguishable from the nuclear genome, while minimizing fluorescence from unbound/non-specifically bound MBs (*Figure 1a*, bottom).

In STORM imaging, the 5 to 10 kb genomic regions labeled with hundreds of oligopaints are usually no more than 500 nm in size (*Boettiger et al., 2016*; *Beliveau et al., 2015*). For the detection of shorter 2 to 3 kb sequences using our method, we expect these nanoscale structures to be even smaller in size. These nanostructures could be mistaken for artifacts such as from insufficient washing during the FISH steps, even though the MB hairpin structure is supposed to eliminate most of the non-specific fluorescence. To address this issue, a blank cell containing the same genomic background without the targeted sequence was used as the absolute negative control. The super-resolution imaging of the target DNA was only considered robust when none or an extremely small amount of nanostructure candidates were detected in the control blank cells. We infected human SK-N-SH cells with a modified lentivirus (*Zufferey et al., 1998*), through which a 3.3 kb viral DNA sequence could be randomly integrated into the nuclear genome. We selected a 2.5 kb fragment within the inserted region as our target sequence (*Figure 1b*). The target sequence contained a CMV promoter and an *egfp* gene, which was driven to express EGFP after viral entry. The EGFP-positive cells (EGFP cells) were sorted and cells carrying the integrated viral DNA in the nuclear genome were used only if they were able to maintain EGFP expression after multiple culture passages. Most of these sorted cells (92.9 ± 1.5%; right peak, solid line) remained fluorescent green after eight further culture passages (*Figure 1—figure supplement 1a*). We performed PCR using genomic DNA extracted from the sorted cells after passaging to confirm the integration of viral DNA. We detected

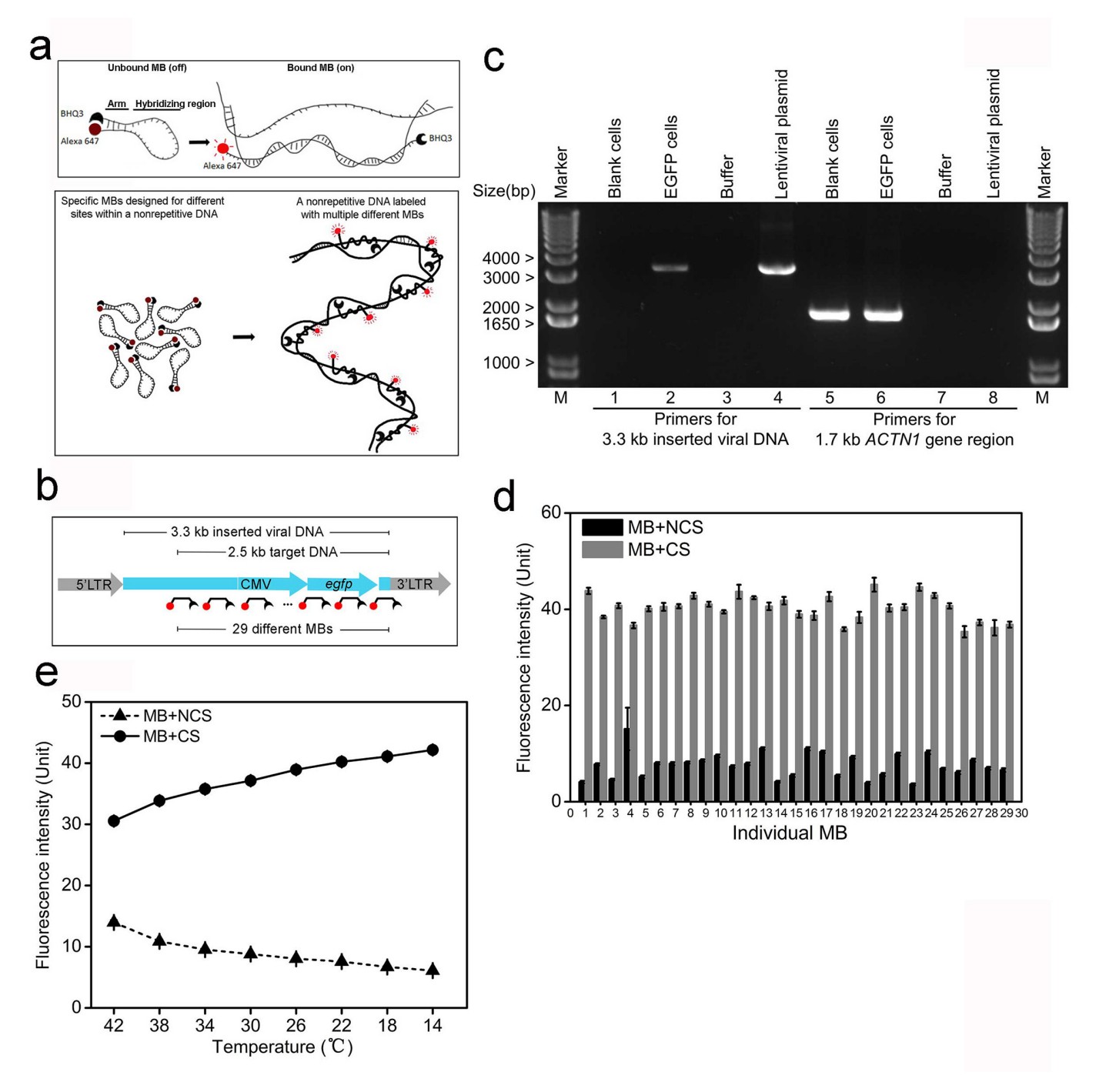

**Figure 1.** MB probes can efficiently reduce non-specific fluorescence and facilitate probe-target binding. (a) Schematic illustration of the MB design for minimizing fluorescence of unbound/non-specifically bound probes. Alexa-647 and BHQ3 was conjugated to MB at 5' and 3' ends, respectively. Two short arms flanking the 42 bp hybridizing region are complementary and will bind to each other in the absence of a complementary sequence (CS) forming a hairpin structure that quenches the fluorescence (top). The set of MB probes were designed to tile along the non-repetitive target DNA in the nuclear genome and fluoresce only upon *in situ* hybridization to the target, which minimizes non-specific fluorescence (bottom). (b) Schematic illustration of the integrated viral DNA with the target in the inserted region. A 3.3 kb non-repetitive lentiviral region (blue) between 5' and 3' LTRs was randomly inserted into the human genome by viral infection. The target DNA was a 2.5 kb sequence containing a CMV promoter and an *egfp* gene within the 3.3 kb inserted region, which was then labeled with the 29 specific MB probes. Each MB is shown as broken-line with red dot (dye) and black crescent (quencher). (c) PCR confirmation of lentiviral integration in the human genome. Using primers targeting the 3.3 kb inserted lentiviral regions (lanes 1–4), a 3.3 kb electrophoretic band was amplified from the lentiviral plasmid (lane 4) and genomic DNA of EGFP cells (lane 2), but not from blank

*Figure 1 continued on next page*

*Figure 1 continued*

controls (lane 1) or PCR mixture without any template (lane 3). Using primers targeting a 1.7 kb portion of human *ACTN1* gene (lanes 5–8), a 1.7 kb PCR product was amplified from genomic DNA of both cells (lanes 5 and 6). Lane Marker: different-sized (bp) DNA ladder bands are shown on the left of gel picture. (**d**) Fluorescence spectrophotometry measurements of 29 individual MB probes (numbered 1–29 in the x-axis) in FISH hybridization buffer with excessive amounts of the corresponding CS (gray bars) or NCSs (black bars) at room temperature. Representative results are shown from three independent experiments. Error bars, SEM. CS: complementary sequence, NCSs: non-complementary sequences. (**e**) Fluorescence spectrophotometry measurements of 29 individual MB probes in the FISH hybridization buffer with excessive amounts of the corresponding CS (solid line with circles) or NCSs (dashed lines with triangles) at different temperatures. Averaged fluorescence readings of the whole probe set are presented for each temperature decreasing from 42°C to 14°C (x-axis). Representative results are shown from three independent experiments. Error bars, SEM. CS: complementary sequence, NCSs: non-complementary sequences.

The following source data and figure supplements are available for figure 1:

**Source data 1.** Design of 29 specific MBs (Viral_MBs) for labeling the 2.5 kb integrated lentiviral target sequence.
**Source data 2.** Source data for 1d and e.
**Figure supplement 1.** EGFP expression in sorted cells after multiple culture passages.
**Figure supplement 2.** PCR confirmation of no tandem repeats of lentiviral integration in a single inserted locus in EGFP cells.
**Figure supplement 3.** Sequence of inserted viral DNA with 29 sites for specific MB probes (Viral_MBs) within the 2.5 kb target region.

a ~3.3 kb band in EGFP cell samples (lane 2) but not in the blank controls (lane 1), whereas both samples showed the 1.7 kb fragment of the human *ACTN1* (lanes 5 and 6) gene (*Figure 1c*). No larger bands of multiples of 3.3 kb were observed in EGFP cell samples (lane 2), indicating no tandem repeats in a single inserted locus. It was also confirmed by an additional PCR test (*Figure 1— figure supplement 2d*). Therefore, we used EGFP cells containing a 2.5 kb unique viral DNA sequence at different genomic loci as well as blank cells (negative control) to establish our *in situ* super-resolution imaging method.

We designed 29 different MB probes for tiling along the 2.5 kb target sequence (*Figure 1—figure supplement 3*). Each MB probe contained a 42-nucleotide (nt) hybridizing region for specific binding to sites along the target strand, as well as two flanking 7-nt arm regions that formed the hairpin structure (*Figure 1—source data 1*). We measured the fluorescence of individual MBs in the presence of excessive amounts of their corresponding complementary sequences (CSs) in FISH hybridization buffer. For each MB, CSs of the other 28 MBs were mixed as the non-complementary sequences (NCSs sharing 10 ± 5 nt complementarity with each MB) and added into the negative control reaction, which represented possible MB fluorescence due to non-specific binding. The spectrophotometry analysis showed high fluorescence intensity for MB probes in the presence of corresponding CS (gray bars), but low fluorescence (19.0 ± 7.1% when normalized to that from CS group) in the presence of NCSs (black bars, *Figure 1d*). This data suggested each unbound/non-specifically associated MB probe had low potential fluorescence of less than a quarter even in the presence of excessive amounts of NCSs sharing 10 ± 5 nt complementarity.

To diminish off-target binding when labeling a unique target, the hybridization step of the oligonucleotide probes in FISH has usually been performed at a relatively high temperature (ranging from 37°C to 47°C) (*Beliveau et al., 2015, 2012*; *Yamada et al., 2011*; *Boyle et al., 2011*; *Boettiger et al., 2016*). To determine an optimal temperature for MB-FISH, we measured the fluorescence of individual MB binding in FISH hybridization buffer at a series of temperatures and presented the averaged fluorescence of the whole MB set for each temperature. As expected, when the hybridization temperature decreased from 42°C to 14°C, MB-CS binding increased (*Figure 1e*, solid line with circles). Surprisingly, the binding of MBs to NCSs gradually diminished with lower temperature (dashed lines with triangles), probably due to the hairpin structure of the probe preventing it from associating with NCSs. Our results suggested the optimal temperature range for MB binding to its single stranded target site was 22°C to 14°C, which was used to determine the final temperature for MB hybridization.

## Optimizing the imaging conditions enables STORM to efficiently distinguish Alexa-647 from the cellular background

To image Alexa-647-labeled genomic sequence(s) at high resolution, we used 3D-STORM with 641 nm and 405 nm laser excitation. The 641 nm laser was used to excite Alexa-647 before switching it back to its dark state. The 405 nm laser was used to reactivate the fluorophore to its fluorescent state, which increases the total number of localizations collected (*van de Linde et al., 2011*; *Fölling et al., 2008*; *Jones et al., 2011*). Each switch from the fluorescent state to the dark state was regarded as a switching event and identified as one localization of the fluorophore. Because a non-repetitive genomic sequence is labeled at a much lower density than a repetitive sequence, Oligopaint-FISH requires an Alexa-405 activator to pair and assist the Cy5 to ensure detection of the fine-scale nanostructure of Cy5-labeled DNA (*Boettiger et al., 2016*; *Beliveau et al., 2015*). Our recent work suggested that sparsely distributed Alexa-647 could be efficiently distinguished from the cellular background even in the absence of an activator dye by optimizing the conditions of STORM imaging. To test this possibility, we used cell samples with relatively uniformly distributed dyes in the entire nucleus rather than MB-hybridized samples, in which fluorescent signals are located only in some local nuclear compartments and are not suitable for comparing multiple conditions. To obtain samples with relatively uniform distributions of fluorophores, we incubated SK-N-SH cells with Alexa-647-conjugated 10T, an oligonucleotide that mediates the non-specific linking of dyes into the nuclear genome.

With the use of these cell samples, we systematically optimized the conditions for STORM imaging. First, we compared the number of events identified under a combination of 12 conditions when imaging the cell samples with the sparsely distributed Alexa-647-conjugated 10T or 10T oligonucleotide as the control (input concentration of 200 pM). The representative image demonstrates the identified events (green dots) in a nuclear area of 14 μm x 14 μm excited under condition XII (*Figure 2a*, left). Each cluster of green dots revealed multiple events during the whole imaging process, likely contributed by one single fluorophore. The zoomed pictures show the sparse distribution of fluorescent dyes in the nuclear field (*Figure 2a*, middle and right). We calculated the event number per unit area for each condition during the 180 s acquisition time (represented as a bar). The results from the control cell (*Figure 2b*, top, white bars) showed there were many switching events observed at a frame rate of 20 Hz (conditions I to IV, 5.3 to 6.9 events per $\mu m^2$), which could be largely reduced by increasing the frame rate to 85 Hz (conditions IX to XII, 0.3 to 2.2 events per $\mu m^2$). These identified events in the control samples were not from 10T oligonucleotides but from cellular autofluorescence, as they were similarly observed in cells with or without oligonucleotide treatment (data not shown). Then, we calculated the false discovery rate (FDR) for each condition, because false fluorescence signals were detrimental to obtaining the real nanostructure of sparsely-labelled non-repetitive DNA. Although various laser power densities and frame rates have been used in other studies (*Doksani et al., 2013*; *Huang et al., 2008b*; *Olivier et al., 2013*; *Rust et al., 2006*; *van de Linde et al., 2011*), we observed the lowest FDR from condition XII (85 Hz, 18 W/cm² 405 nm laser and 1 kW/cm² 641 nm laser; marked with asterisk in *Figure 2b*), which was the optimal condition for imaging sparsely distributed Alexa-647.

During STORM data acquisition, the imaging buffer/environment becomes acidic and less capable of removing oxygen, which in turn suppresses the switching properties of fluorophores (*Vogelsang et al., 2009*; *Aitken et al., 2008*; *Dempsey et al., 2011*). To test if condition XII was still the optimal condition in the changing sample environment during acquisition, we modified the standard imaging buffer from pH 8.0 to a relatively lower pH of 7.5 and reduced the concentration of glucose oxidase and catalase by half. In this non-ideal imaging environment, condition XII still had the lowest FDR among all analyzed conditions (*Figure 2c*), indicating it was the most optimal condition.

Next, we assessed the possible sample photo-bleaching effects (*Annibale et al., 2011*; *Fölling et al., 2008*) of condition XII compared to three other conditions (IX, X, and XI), which had the same frame rate of 85 Hz but different laser powers. We calculated the event numbers every 5 s in each entire sequence of images and presented them as values normalized to the first 5 s event number. Although laser powers differed across conditions, we found the normalized event numbers all decreased quickly during the first 90 s and did not vary much in the last 90 s (*Figure 2d*), which indicated a dark-fluorescent quasi-equilibrium with time (*Vogelsang et al., 2009*; *Dempsey et al.,*

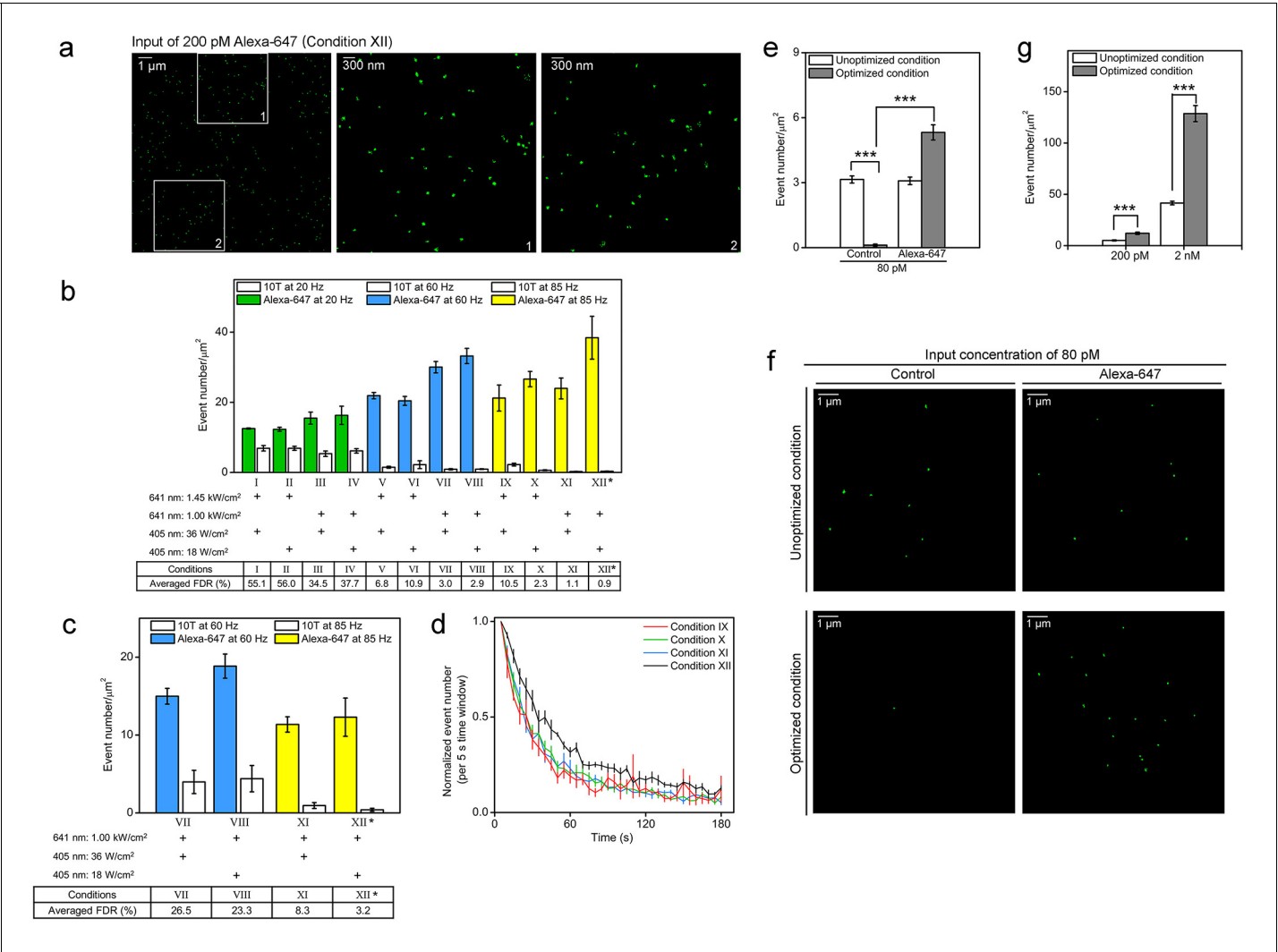

**Figure 2.** Optimized STORM imaging conditions allow efficient identification of Alexa-647 from the background cellular autofluorescence. (**a,b**) Nuclei of fixed SK-N-SH cells were sparsely labeled with Alexa-647-conjugated 10T or 10T controls (input concentration of 200 pM) via non-specific association between oligonucleotide and the nuclear genome at low temperature (4°C). Randomly selected fields were imaged in standard imaging buffer (pH 8.0) using 3D-STORM under the indicated conditions for 180 s. (**a**) Representative STORM images show switching events (green dots) within a 14 μm x 14 μm nuclear area excited under condition XII (left panel). The middle and right panels are the zoomed views of the two white boxed regions (1 and 2) in the left panel, respectively. The green dots show the fitted position of each event. Each cluster of green dots indicates multiple events collected during the whole imaging process, probably contributed by a single dye. The pseudo-color (green) is used to visualize the dots. (**b**) Event number per unit area (event number/μm$^2$) is shown as a bar, representing the results from Alexa-647-conjugated 10T (color) or control (white) samples. Power density of the 641 nm or 405 nm laser is shown below each bar. False discovery rate (FDR) calculated as the ratio of event number per μm$^2$ from controls to that from cells treated with Alexa-647-conjugated 10T is shown at the bottom. (*) indicates the condition with lowest FDR value. Representative results are shown from three independent experiments. Error bars, SEM. Control, 10T; Alexa-647, Alexa-647-conjugated 10T. (**c**) Cell samples were labeled with Alexa-647-conjugated 10T or 10T controls (input concentration of 200 pM) and imaged in a modified buffer containing half concentrations of glucose oxidase and catalase (GLOX) at a relatively low pH of 7.5. Conditions VII-VIII at 60 Hz and XI-XII at 85 Hz were used in the assessments. (*) indicates the condition with lowest FDR value. (**d**) Event numbers detected per 5 s (*Figure 2b*) under experimental conditions IX to XII at 85 Hz were calculated and normalized to the first time slot. The plot shows the trend during the entire image series. Error bars, SEM. (**e,f**) Cell samples were labeled with Alexa-647-conjugated 10T or 10T controls (input concentration of 80 pM) and imaged by optimized or unoptimized STORM (conditions I and XII in *Figure 2b*, respectively). (**e**) Event numbers per unit area (event number/μm$^2$) are shown as gray and white bars, representing data collected under optimized and unoptimized conditions, respectively. Representative results are shown from three independent experiments. Error bars, SEM. ***p<0.001; t test. (**f**) Representative STORM images showing switching events (green dots) in 10 μm x 10 μm nuclear areas excited under optimized (top) or unoptimized (bottom) conditions. The green dots indicate the fitted position of each event. Each cluster of green dots indicates multiple events during the whole imaging process, probably contributed by a single dye. The pseudo-color (green) is used to visualize the dots. (**g**) Cell samples were labeled with Alexa-647-conjugated 10T or 10T controls at 200 pM or 2 nM input concentration and imaged by optimized (gray) or unoptimized (white) STORM

*Figure 2 continued on next page*

*Figure 2 continued*

(Conditions I or XII in *Figure 2b*, respectively). Event numbers per unit area (event number/µm²) from Alexa-647-labeled cells were subtracted from controls (data not shown) and are represented as bars. Representative results are shown from three independent experiments. Error bars, SEM. ***$p<0.001$; t test.

The following source data is available for figure 2:

**Source data 1.** Source data for 2b, c, d, e and g.

*2009*). The highest proportion of events during the whole imaging process still occurred under the optimized condition (dark line) compared to the other conditions (*Figure 2d*, colored lines), indicating less photo-bleaching of Alexa-647 in optimized condition XII.

Because a sufficient number of localizations are required to obtain fine-scale DNA nanostructures, we next evaluated the detection efficiency of Alexa-647 under optimized (XII) or unoptimized (I) conditions, the latter was comparable to the condition used in a previous report (*Huang et al., 2008b*). In cells with extremely sparse distributions of Alexa-647 (input concentration of 80 pM), fluorescence signals could be efficiently identified only under optimized conditions, likely due to differences in autofluorescence of the control samples between unoptimized and optimized conditions ($3.15 \pm 0.17$ vs. $0.12 \pm 0.05$ events per µm²; *Figure 2e*). This indicated the optimized condition was more suitable for very sparsely labeled cases, although conditions comparable to the unoptimized conditions could work in densely labeled targets (*Huang et al., 2008a*, *2008b*). Representative images of extremely sparse distributions of Alexa-647 showed more events identified under optimized conditions (*Figure 2f*, right bottom panel) and relatively high autofluorescence observed in control cells under unoptimized conditions (*Figure 2f*, left top panel). In samples with relatively high Alexa-647 density (input concentration of 200 pM or 2 nM), we observed more switching events using the optimized conditions (gray bars) compared to unoptimized conditions (white bars) after subtracting the corresponding autofluorescence events (*Figure 2g*). Taken together, condition XII was considered to be the optimal condition for non-repetitive DNA imaging based on the low FDR, minimal photo-bleaching, and higher efficiency of Alexa-647 detection.

## Resolution of 3D-STORM using optimized conditions for sparsely distributed Alexa-647

In this study, 3D-STORM was used to enable high-precision localization of the otherwise spatially overlapping images of individual Alexa-647 on MB-labeled target DNA. However, the resolution of STORM is limited by the number of photons and localization numbers (*van de Linde et al., 2011*; *Thompson et al., 2002*; *Deschout et al., 2014*; *Huang et al., 2010*) and hence is affected by fluorophore properties and imaging conditions. We determined the localization precision of 3D-STORM imaging in cells sparsely labeled with Alexa-647 under optimized condition. Alexa-647 within the illuminated nuclear fields was switched on and off over multiple cycles to obtain a cluster of localizations from each fluorophore. We aligned 1378 localizations from 53 clusters by their centroids to give an overall 3D distribution (*Figure 3a*). The standard deviations of the distribution of these localizations were 9 nm in lateral and 22 nm in axial dimensions, and the corresponding full width at half maximum (FWHM) values were 22 nm and 52 nm, respectively (*Figure 3b*), which yielded a resolution of 20–30 nm in x-y and 50–60 nm in z directions. *Figure 3c* shows an example of two clusters of localizations separated by 95 nm in the z axis being well resolved according to the z profile.

A 2.5 kb genomic sequence can form 10 nm nucleosomes by wrapping around a histone complex (approximately 200 bp sequence on one nucleosome) or even fold into a compacted structure (*Luger et al., 2012*; *Bartholomew, 2014*; *Schones et al., 2008*; *Luger et al., 1997*; *Schmitt et al., 2016*). Therefore, we expect our MB-labeled target to be detected as very tiny structures in STORM, possibly up to 300 nm in size when referred to the sizes of Oligopaint-labeled DNA nanostructures (*Boettiger et al., 2016*; *Beliveau et al., 2015*). Such nanostructures could be mistaken for artifacts due to sample stage drift. We used fluorescent beads as fiducial markers to correct for sample drift as previously described (*Rust et al., 2006*). The beads were added to the cell samples and imaged together with the MB-labeled target(s) in the same field. *Figure 3d* shows the distributions of 492 localizations from 23 Alexa-647 fluorophores were less spread out after stage drift correction. Thus,

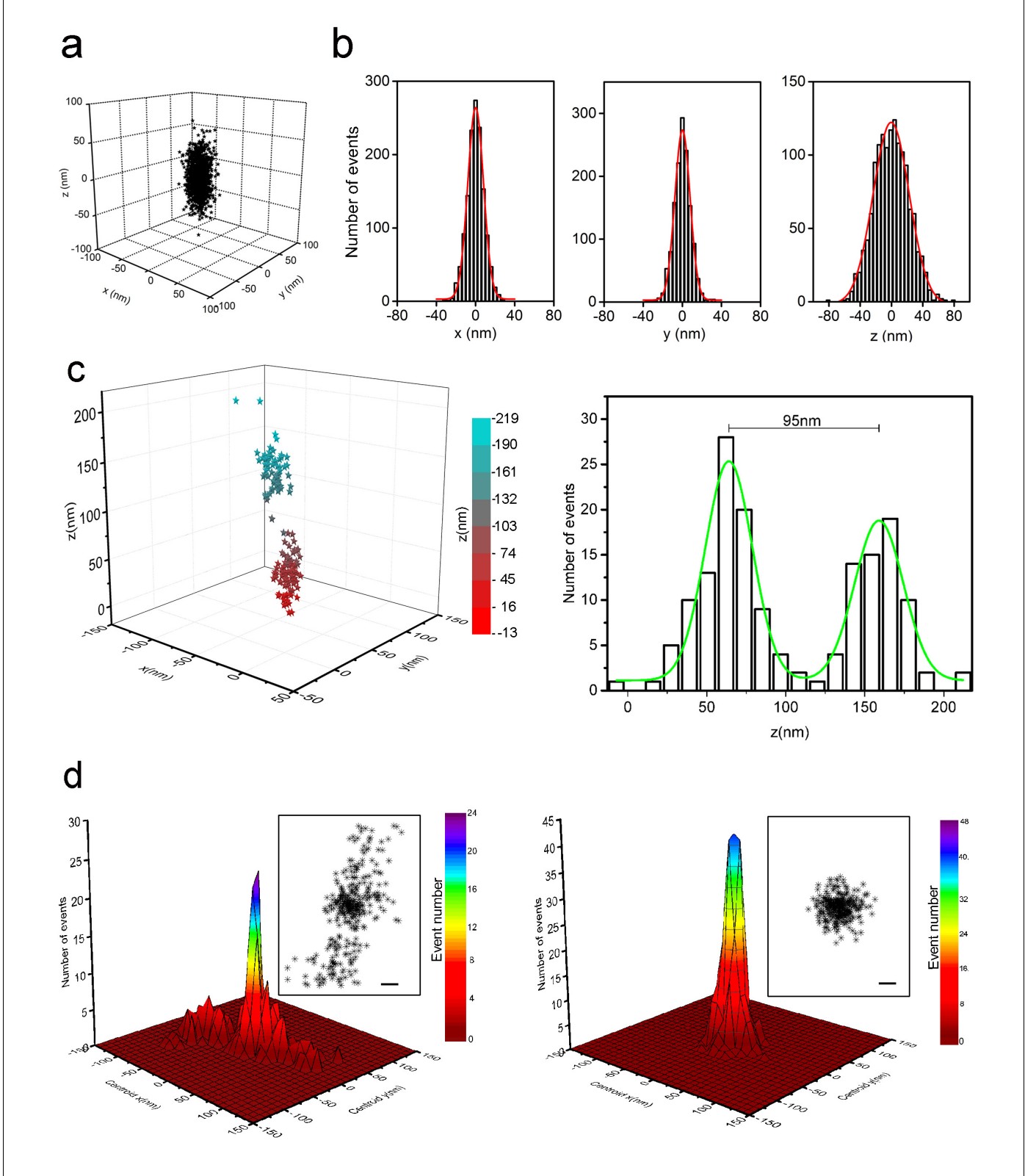

**Figure 3.** Localization precision of 3D-STORM for sparsely distributed Alexa-647 in cells with refined optical setup. (**a**) 3D distribution of localizations from individual Alexa-647 fluorophores in fixed SK-N-SH cells under optimized conditions. Cells were sparsely labeled with Alexa-647 molecules (input concentration of 80–200 pM). Each Alexa-647 fluorophore contributed a cluster of localizations from a series of fluorescent/dark switching cycles. The 1378 localizations from 53 clusters were aligned by their centroid positions to generate the overall 3D presentation of the distribution of localizations. *Figure 3 continued on next page*

*Figure 3 continued*

(**b**) Histograms of the 1378 localizations were fitted to a Gaussian function yielding standard deviations of 9 nm in x, 9 nm in y, and 22 nm in z axes. The corresponding FWHM values were 22 nm in x, 22 nm in y, and 52 nm in z axes. (**c**) Two neighboring clusters of 160 localizations from cells sparsely labeled with Alexa-647. The 3D distribution of localizations is represented as color-coded z-axis information (left, −13 nm to 219 nm). The histogram shows the z coordinate distribution of these localizations fitted into two Gaussian curves with a separation of 95 nm between the two peaks (green curve, right). (**d**) Localization precision of the STORM imaging before (left) and after (right) stage drift correction in cells with sparsely distributed Alexa-647, which shows 492 localizations collected from 23 fluorophores that blinked more than 10 times during the entire imaging process. Their lateral positions were realigned so that the average centroid position was at the origin. The main plots show histograms of these centroid positions. The overall distribution of all localizations is shown in the insets. Scale bars are 20 nm. The drift-corrected distribution (right) became significantly narrow compared to the uncorrected distribution (left).

drift correction using fluorescent beads will be applied when imaging MB-labeled sequences and reconstructing super-resolution images.

## Super-resolution visualization of 2.5 kb non-repetitive DNA *in situ* in the human genome

To label the 2.5 kb unique viral DNA sequence integrated in human genome, we simultaneously subjected EGFP or blank SK-N-SH cells to MB-FISH using the 29 probes after confirming EGFP expression in those cells (*Figure 1—figure supplement 1b*). Fluorescent beads were added to cell samples for stage drift correction immediately before the imaging experiments. The nuclear field was selected under bright field and captured by conventional microscopy using two different lasers separately to discriminate the putative MB-labeled target(s) visible only at 641 nm laser excitation from the fluorescent beads visible stably at 405 nm and occasionally at 641 nm due to weak excitation and heterogeneous bead properties. The dots co-appearing at the same positions excited by the 405 nm (row B) or 641 nm (row C) lasers were from fluorescent beads (shown by yellow arrows in Cell II in *Figure 4a*). Multiple sequential images, each containing 18000 frames, were collected from one layer (700 nm in depth) in each nucleus. We reconstructed the sequential frames into a super-resolution image after drift correction. In different EGFP cell nuclei (outlined in red in *Figure 4a*, top row), we consistently obtained specific fine-scale nanostructures (*Figure 4a*, row D) of MB-labeled target DNA, shown as the detailed morphologies in the zoomed views (*Figure 4a*, row E). These structures identified in the STORM imaging appeared as dots (Cell I and III) or occasionally as unrecognizable structures (Cell II) in conventional imaging using weak 641 nm laser, likely due to varied amounts of MBs on the target sequence. Furthermore, the positions of these nanostructures did not overlap with the beads visible under 405 nm laser excitation (*Figure 4*, row B), which excludes the possibility that the identified nanostructures result from interference from bead emissions. The z information, represented as pseudo-colors, ranged from −350 nm (purple) to +350 nm (red) in the z-axis as indicated by the color bar beneath the images. These images were composed of 100–1800 localizations, which is comparable to the number of localizations (100–400) collected from the Oligopaint-labelled 4.9 kb sequences (*Beliveau et al., 2015*) and sufficient for delineating fine-scale 3D structures.

We rarely found complex structural features in the nuclear fields of the blank controls, rather we observed sparsely distributed clusters of localizations, such as those in the right panel of *Figure 4b*. These relatively sparse signals were also detected in EGFP cell nuclei (*Figure 4b*, left panel), possibly reflecting unbound/non-specific probes. As shown in *Figure 4c*, the corresponding FWHMs along cross-sections defined by the red line in *Figure 4a* showed our method was able to resolve substructures separated by a lateral distance of 41 nm or structural features of 35–43 nm in size (*Figure 4c*). Collectively, we detected nanostructures in 21 out of 78 EGFP cells across multiple independent experiments, but zero in the 83 blank cells (hypergeometric p value=5.0 × 10$^{-8}$). These findings demonstrate the capability and specificity of our MB-FISH imaging method.

## Super-resolution visualization of 2.5 kb enhancer *in situ* in *Nanog* locus of mouse ESCs

Next, we test whether our MB-FISH method could also be used to visualize an endogenous target, especially those non-coding elements like enhancers or promoters. As a secondary target for super-

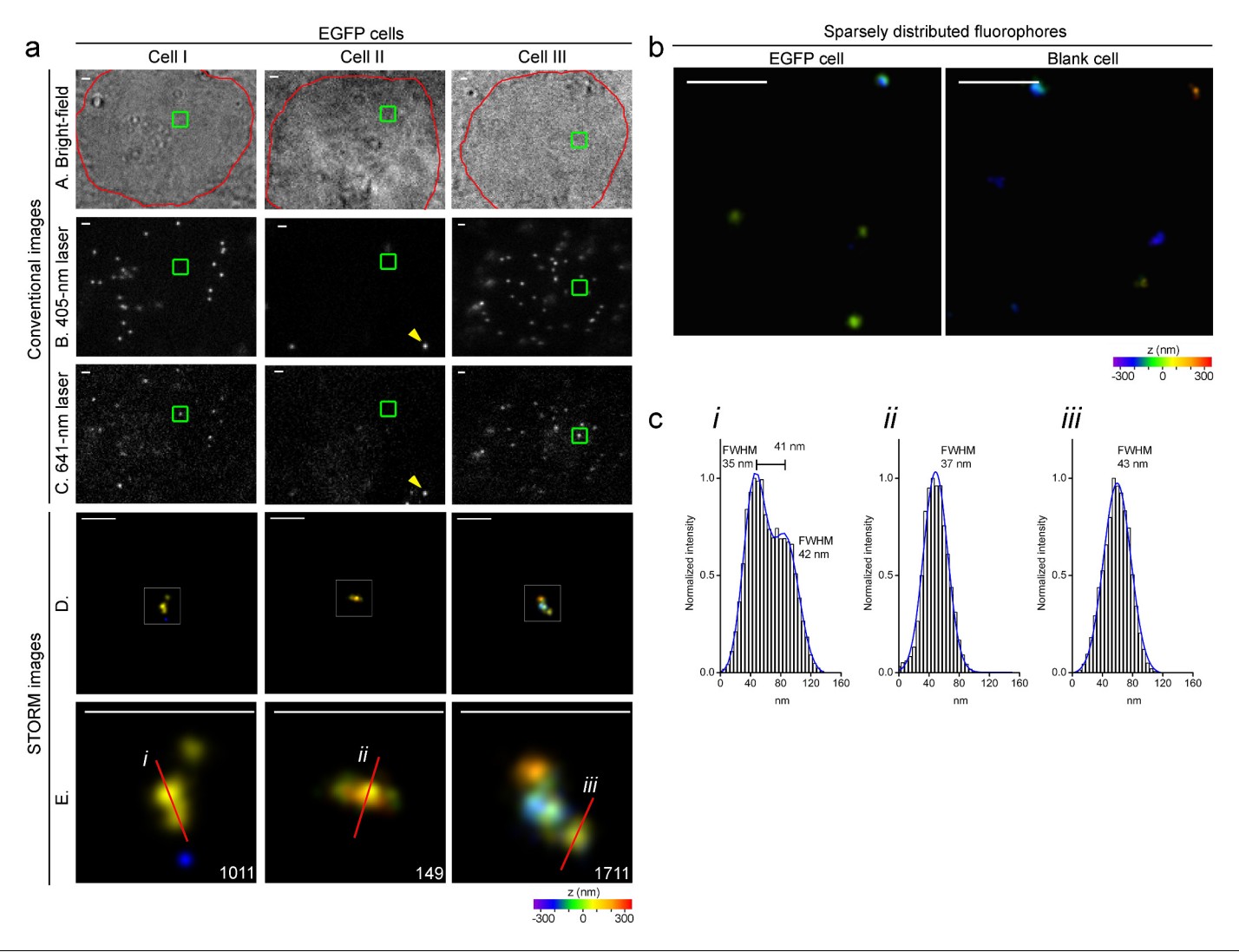

**Figure 4.** Specific nanostructures of the 2.5 kb integrated DNA labeled with MBs *in situ* in the human genome. (a) Representative specific nanostructures of MB-labeled 2.5 kb integrated lentiviral DNA in EGFP cells (Cell I to III). The conventional view of nuclear region in bright field (outlined in red) or excitation by 405 nm or 641 nm laser are shown in top three rows (rows A, B, and C), respectively. The nanostructures were identified from the drift corrected STORM images corresponding to the green box region in conventional images) (row D). Zoomed views of the white boxed regions in row D reveal more detailed morphology of the detected 3D structures (bottom row), and the localization number of each nanostructure is shown in the lower-right corner. Each localization number represents the detected times of the Alexa-647 fluorophores labeled on DNA during the entire imaging process. STORM images are shown as rainbow color-coded z-axis information (color bar at the bottom, −350 to 350 nm). A representative fluorescent bead that emits under both 405 nm and 641 nm laser excitation is highlighted by yellow arrows in panel B and C of Cell II. Scale bars are 1 μm in top three rows and 300 nm in bottom two rows. (b) Representative discrete signals observed in the STORM images of MB-labeled EGFP (left) or blank (right) cell nuclei, possibly from unbound/non-specific probes. The STORM images are shown after drift-correction with rainbow color-coded z-axis information (color bar at the bottom, −350 to 350 nm). Scale bars are 300 nm. (c) Histograms of the normalized number of counts detected (Normalized intensity) along the cross-sections defined by the red lines (i-iii) of the STORM images in (a). Values of FWHM indicate the features above each structure along the transverse positions with 1D Gaussian fit.

resolution visualization, we chose a 2.5 kb super-enhancer at −45 kb upstream within the *Nanog* locus in mouse embryonic stem cells (mESCs) (***Blinka et al., 2016***). Avoiding repeats that appear in other genomic regions, we designed 34 MB probes according to the target sequence: 24 MBs for minus strand and 10 MBs for plus strand (***Figure 5—figure supplement 2*** and ***Figure 5—source data 1***). The spectrophotometry analysis revealed that these MBs had dramatically reduced background fluorescence (15.8 ± 5.9%) in FISH hybridization buffer in the presence of NCSs, compared

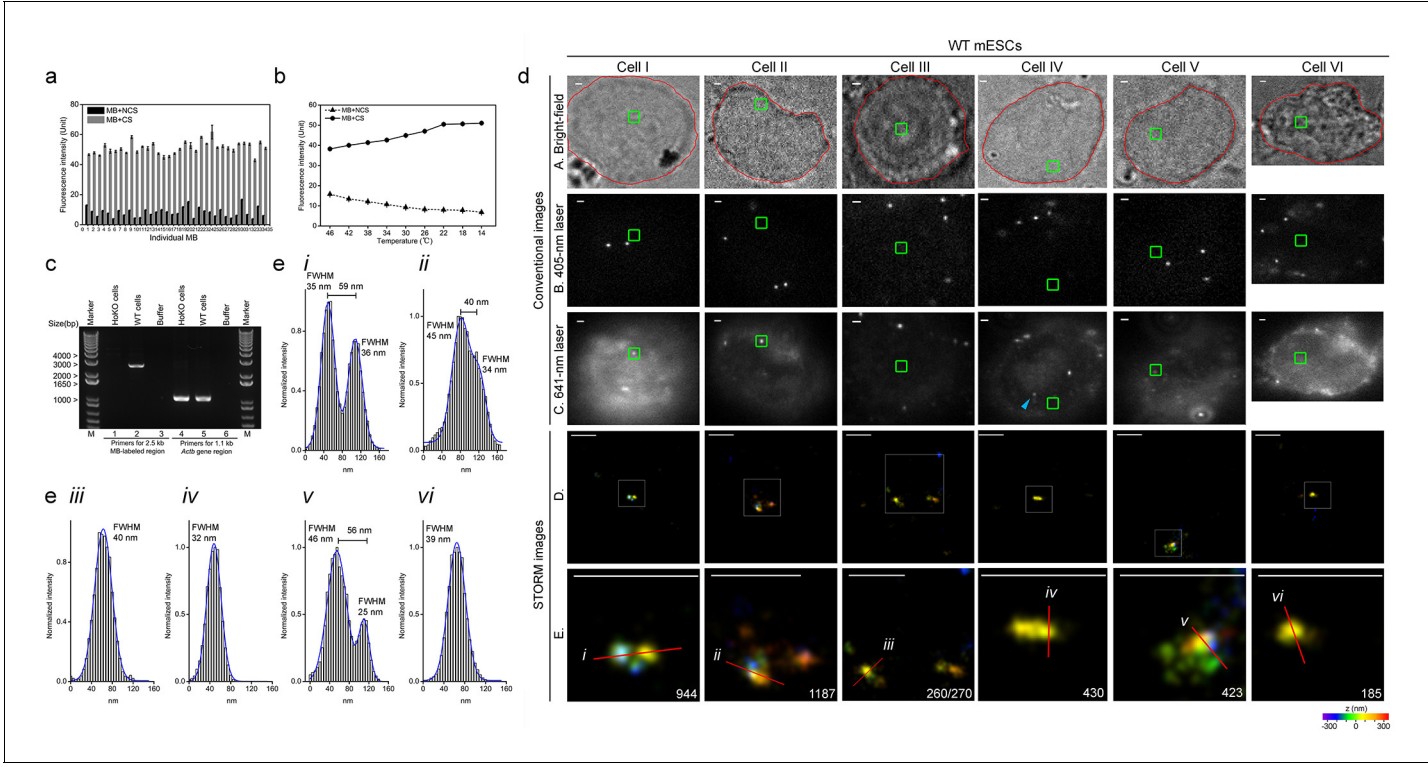

**Figure 5.** Specific nanostructures of 2.5 kb MB-labeled *Nanog* enhancer in mESC nuclei. (**a**) Fluorescence spectrophotometry measurements of 34 individual Nanog_MB probes (numbered 1–34 in the x-axis) in FISH hybridization buffer with excessive amounts of the corresponding CS (gray bars) or NCSs (black bars) at room temperature. Representative results show the inhibition of non-specific fluorescence was 15.8 ± 5.9% compared to the fluorescence reading in the presence of CS. Error bars, SEM. CS: complementary sequence, NCSs: non-complementary sequences. (**b**) Fluorescence spectrophotometry measurements of 34 individual Nanog_MB probes in the FISH hybridization buffer with excessive amounts of the corresponding CS (solid line with circles) or NCSs (dashed lines with triangles) at different temperatures. Averaged fluorescence readings of the whole probe set are presented for each temperature decreasing from 46°C to 14°C (x-axis). Representative results are shown from three independent experiments. Error bars, SEM. CS: complementary sequence, NCSs: non-complementary sequences. (**c**) PCR confirmation of homozygous knockout of MB-labeled *Nanog* enhancer from both alleles. Using primers targeting the endogenous 2.5 kb Nanog_MBs target region (lanes 1–3), a 2.5 kb electrophoretic band was amplified from genomic DNA of WT mESCs (lane 2) but not from HoKO mESCs (lane 1) or PCR mixture without any template (lane 3). Using primers targeting a 1.1 kb portion of mouse *Actb* gene (lane 4–6), a 1.1 kb PCR product was amplified from genomic DNA in both cells (lane 4 and 5). Lane Marker: the different sized (bp) DNA ladder bands are shown on the left of gel picture. (**d**) Representative specific nanostructures of MB-labeled 2.5 kb endogenous enhancer in *Nanog* locus in CJ9 mESCs (Cell I to VI). The conventional view of a cellular region in bright field (outlined in red) or excited by 405 nm or 641 nm lasers are shown in the top three rows (rows A, B, and C), respectively. The nanostructures were identified from the drift corrected STORM images corresponding to the green box region in conventional images (row D). Zoomed views of the white boxed regions in row D reveal more detailed morphology of the detected 3D structures (bottom row) with the localization number of each nanostructure shown in the lower-right corner. Each localization number represents the detected times of the Alexa-647 fluorophores labeled on DNA during the entire imaging process. STORM images are shown with rainbow color-coded z-axis information (color bar at the bottom, −350 to 350 nm). Three representative fluorescent dots visible under 641 nm laser excitation but not identified as specific nanostructures in STORM reconstruction are highlighted by blue arrows in panel C of Cell III. Scale bars are 1 μm in top three rows and 300 nm in bottom two rows. (**e**) Histograms of the normalized number of counts detected (Normalized intensity) along the cross-sections defined by the red lines (i–vi) in the STORM images in (**d**). Values of FWHM indicate the features above each structure along the transverse positions with 1D Gaussian fit.

The following source data and figure supplements are available for figure 5:

**Source data 1.** Design of 34 specific MBs (Nanog_MBs) for labeling the 2.5 kb *Nanog* enhancer in Nanog locus in mESC nuclei.

**Source data 2.** Source data for 5a and b.

**Figure supplement 1.** Identification of nuclear periphery of mESCs by conventional co-imaging using bright field and DAPI staining.

**Figure supplement 2.** Sequence of endogenous DNA with 34 sites for specific MB probes (Nanog_MBs) within the 2.5 kb target region.

*Figure 5 continued on next page*

*Figure 5 continued*

**Figure supplement 3.** Sequence information of allele identified in homozygous knockout (HoKO) ESC clone.
**Figure supplement 4.** A representative of identified nanostructure or excluded fluorescence noise and their corresponding appearances in 641 nm conventional imaging.

to the fluorescence readings in the presence of the CS (*Figure 5a*). We also observed an obvious trend of increased specific and decreased non-specific binding with lowering hybridization temperatures (*Figure 5b*), which also confirmed the optimal temperature range for MB-CS binding was 22°C to 14°C. To obtain a cell without this super-enhancer as the negative control, we applied CRISPR/Cas9-mediated knockout in CJ9 mESCs (*Shen et al., 2013*; *Yin et al., 2015*) to generate homozygous knockout (HoKO) cells, in which a 3 kb region covering the super-enhancer target was deleted from both alleles. The knockout was confirmed by PCR (*Figure 5c*) and sequencing (*Figure 5—figure supplement 3*).

After sub-culturing for 18–24 hr, both wild-type (WT) and HoKO ESCs without typical colony morphology were simultaneously subjected to MB-FISH followed by 3D-STORM imaging. Most of these cells were relatively round and weakly adherent. Their nuclear peripheries were not recognizable under bright field microscopy. To determine the position of localization and tell if a potential nanostructure sits within nucleus, we performed co-imaging of DAPI and bright field and found the nuclear peripheries very close to the cell outlines (*Figure 5—figure supplement 1*). A nuclear field was selected and subjected to conventional imaging using two different lasers separately to discriminate the fluorescent beads from putative MB-labeled target(s). Although the fluorescence background from conventional 641 nm laser imaging (*Figure 5d*, row C) varied among cells across multiple experiments, nanostructures were repeatedly observed in WT cells in STORM images after drift correction (*Figure 5d*, rows D and E). STORM imaging data were collected from a 700-nm-in-depth layer in each nucleus (around 5 μm in depth), leaving most of the nuclear volume unexplored. We detected MB-labeled structure(s) in 14 of the 92 WT cells, but none in any of the 74 HoKO cells (hypergeometric p value=$1.6 \times 10^{-4}$). These results demonstrated the specificity and robustness of our MB-FISH imaging method.

Usually, each cell with positive signals was detected with one nanostructure, but sometimes two structures were detected in a single nucleus (Cell III in *Figure 5d*). The identified structures were composed of 150–1200 localizations (numbers in the bottom row in *Figure 5d*), comparable to numbers observed in the integrated viral DNA imaging. Fluorescent dots (Cell I, II or V in *Figure 5d*, row C) or unrecognizable structures (Cell III, IV or VI in *Figure 5d*, row C) were observed in 641 nm conventional imaging at positions corresponding to the identified nanostructure (highlighted in the green boxes). Occasionally, we identify no potential nanostructure of labeled DNA in the STORM imaging at region corresponding to fluorescent dot visible under 641 nm laser excitation (indicated by blue an arrow in Cell IV in *Figure 5d*, row C). The corresponding FWHMs along cross-sections (indicated by a red line in *Figure 5d*) demonstrated the ability of our method to resolve sub-structures separated by a lateral distance of 59 nm and intriguing structural features of 25–46 nm in size (*Figure 5e*). Taken together, the combination of MB-FISH and 3D-STORM imaging allows super-resolution visualization of 2.5 kb unique genomic elements *in situ* in the nuclear genome.

## Discussion

We demonstrated our MB-FISH method could directly visualize 2.5 kb non-repetitive DNA sequence *in situ* in human or mouse genome, which is the shortest unique genomic sequence resolved to date at super-resolution. Similar to the Oligopaint-labeling method (*Beliveau et al., 2012*; *Boyle et al., 2011*; *Beliveau et al., 2015*; *Yamada et al., 2011*), a set of MB probes can be bioinformatically designed to label a known unique genomic locus on either target strand, avoid repetitive sequences, and can provide allele specificity by probing sites covering single-nucleotide polymorphisms (SNPs) from maternal or parental chromosome. In the future, we expect our MB-FISH could be combined with other super-resolution techniques, such as STED (*Donnert et al., 2007*; *Yang et al., 2016*), only if these techniques must be able to image fluorescent dyes at a density of a few tens of molecules in

2.5 kb genomic DNA. For a good combination, the imaging conditions must be optimized to efficiently distinguish dyes from background cellular autofluorescence. In regard to suitable fluorophores for MB probes, any reported dye used in STORM imaging with an available corresponding quencher could be applicable, such as Atto-488, Atto-565, Alexa-647, or Cy5 (*Dempsey et al., 2011*; *Heilemann et al., 2009*, *2008*; *Lehmann et al., 2015*; *Rust et al., 2006*). It might be possible to label and image three genomic elements using sets of MBs carrying Atto-488, Atto-565, or Alexa-647, respectively, as well as using multi-color STORM to detect interactions among loci *in situ* in single cells. This could be an important approach to shed new light on looping interactions and 3D chromatin organizations in single cells at super-scale resolution.

Oligopaint-FISH is a pioneering method for visualizing genomic loci at the nanoscale resolution and can provide important information for other studies. For example, the demonstration that localization density is essential for a fine-scale structure in Oligopaint-FISH encourages us to optimize STORM imaging conditions and achieve more localizations in our MB-FISH method. Nonetheless, MB-FISH differs from Oligopaint-FISH in several ways. First, to achieve sufficient localization density, Oligopaint-FISH depends on a secondary probe, whereas MB-FISH appeals to imaging condition optimization. Second, both methods involve high concentrations of fluorophore-conjugated probes (0.71 µM in MB-FISH vs >1 µM in Oligopaint-FISH), which unavoidably leads to interference from non-specific fluorescence noise. To address this issue, Oligopaint-FISH applies a relatively high temperature for hybridization (37°C, 42°C or 47°C), whereas MB probes dramatically quenches fluorescence from unbound or non-specifically bound probes (*Figure 1d*). Third, the hybridization temperature used by MB-FISH (22°C) was lower than that used by Oligopaint-FISH (up to 47°C) and chosen based on optimal MB-CS binding, chromatin accessibility as well as simplifying the FISH steps (performed at laboratory temperature). The low hybridization temperature reduces background fluorescence (*Figure 1e*) and promotes probe-target binding, hence decreasing the number of input probes needed for the STORM imaging (as few as 29 probes for MB-FISH vs ~106 probes for Oligopaint-FISH). Fourth, Oligopaint-FISH has a sequence resolution of 4.9 kb, but MB-FISH managed to improve the resolution down to 2.5 kb due to the need of fewer input probes and the lowered melting temperature for the hybridizing region. With the use of MB-FISH, it may be possible to directly visualize transient pairing of narrowed elements within Oct4 enhancer/promoter region during the onset of ESC differentiation (*Hogan et al., 2015*). Fifth, the hybridizing region of Oligopaint was of a length of 32 to 42 nt, whereas MB adopted a hybridizing region length of 42 nt to ensure the probe hairpin structure would break upon associating with target sequence. In the future, we will further optimize the length of hybridizing region for MB-FISH to be even shorter, so that similar numbers of optimized MB probes can be designed within shorter target sequence at higher density than the current sets (11.6 probes/kb or 13.6 probes/kb for exogenous or endogenous experiment, respectively), hence further improving sequence resolution. Sixth, similar to the distance between two neighboring Oligopaints, the nearest distance between two MB sites on a target strand is expected to be around 10 nt. Thus, the proximal distance between an Alexa-647 and a BHQ3 from two neighboring MBs will be 24 nt (10 nt plus the length of two flanking 7-nt arms), which is about 7–8 nm and largely avoids inter-molecular quenching (*Wu and Brand, 1994*). Finally, the large numbers of Oligopaints needed lead to a dependence on a probe generation system in the laboratory, whereas the fewer probes needed in MB-FISH can be obtained commercially, making it relatively simple to be used in the laboratory. However, to label large domains at megabase level, the cost of synthesizing MB probes will become very high, while home-made Oligopaints are more suitable for the task. In this regard, MB-FISH is suitable for shorter elements such as enhancers or promoters, whereas Oligopaint-FISH works well in large chromosome domains (*Beliveau et al., 2015*; *Boettiger et al., 2016*; *Wang et al., 2016*). Taken together, MB-FISH could be a complementary approach to other existing methods, including Oligopaint-FISH, for visualizing a non-repetitive genomic sequence.

After reconstruction of super-resolution images, areas containing more than 100 localizations were saved for further filtering according to the following criteria. First, the DNA nanostructure should be localized within the nucleus. Boundaries around nuclei were defined under bright field illumination. Second, nanostructure candidates overlapping with fluorescent beads were excluded to avoid any artifacts caused by bead emissions. Third, given the multiple MBs labeled on the target DNA, the fluorophores on a nanostructure were expected to blink stochastically and be sparsely activated during the whole imaging process. Localizations crossing a short series of continual frames

during the whole acquisition time have been occasionally observed in negative control samples (blank SK-N-SH cells or HoKO ESC) and filtered out as noise. Fourth, a sufficient localization density was essential to generate a high-quality image of the nanostructure, and those nanostructure candidates not satisfying the localization densities needed were discarded. Fifth, since the number of MB probes on a labeled DNA could be upwards of 29 or 34 in this study, the localization number of a nanostructure was expected to be below a certain value (3000). Those candidates with localization numbers beyond this limit were excluded. Finally, regardless of how the 2.5 kb target DNA might be folded within the genome, its projected lateral dimension must be physically restricted and identifiable, given our 3D-STORM resolution. The selected nanostructures should not exceed the area limit, which was determined to be 0.2 um$^2$ according to the data collected, and candidates with widely spread out were excluded from further analysis. Collectively, these stringent criteria have been empirically determined to minimize false positives, albeit at the expense of false negatives (or true positives), leading to specific identification of MB-labeled nanostructures only in positive cells.

To minimize false-positives in detecting short non-repetitive genomic sequences, we applied very stringent criteria to eliminate noise. As a result, no nanostructures were detectable in the negative controls of blank SK-N-SH cells or HoKO ESCs. Nevertheless, nanostructures were stably detected in positive cells, more EGFP cells (21/78; ~27%) than ESCs (14/92; ~15%) having detectable nanostructures. Higher percentage of cells with nanostructure(s) are expected to be observed by collecting multiple layers from the whole nucleus (at least 5 µm in depth), although only a single nuclear layer (700 nm in depth) was collected each cell in this proof-of-principle study. However, the labelled efficiency could never reach 100%, since those cells undergoing mitosis are insensitive to either Oligopaint or MB labeling due to the formation of highly compacted chromosomes. Higher ratio of ESCs is known to undergo mitosis (*Ahuja et al., 2016*; *White and Dalton, 2005*), possibly leading to the different detection efficiency in ESC and EGFP cells.

Most nanostructures with large amounts of localizations appeared as distinct fluorescent dots in 641 nm conventional imaging (Cells I and II in *Figure 5d*), whereas those with fewer localizations were not visualized as dotted structures (Cell III and VI in *Figure 5d*). Occasionally, nanostructures visible as fluorescent dots in 641 nm conventional imaging did not consist of many localizations (Cell V in *Figure 5d*), likely due to the relatively lower blinking frequency of fluorophores on that nanostructure. Meanwhile, not all fluorescent dots observed in conventional imaging were identifiable as specific nanostructures when subjected to STORM imaging. For example, when reconstructing STORM image of region corresponding to the dot marked by a blue arrow in Cell IV in *Figure 5d* (row C), we observed clusters of sparsely distributed localizations with extremely low number (distance of a few tens of nanometers between clusters composed of less than 100 localizations) (*Figure 5—figure supplement 4*). Such a case was excluded from DNA nanostructure candidates and regarded as non-specific signals according to our stringent criteria in nanostructure identification.

The 2.5 kb MB-labelled DNA was occasionally observed to have folded structure, which was not surprising given the bendability/flexibility of short DNA lengths (*Vafabakhsh and Ha, 2012*). Some of the identified nanostructures consisted of multiple distinct and contiguous dots (e.g., Cell I in *Figure 4a* or *Figure 5d*). These structures could be distinguished from the sparsely distributed signals of non-specific dyes by the blinking pattern along the whole acquisition time, the localization intensity, and the different spreading levels of localizations. Possibly at least two effects contributed to these nanostructures with distinct dots. On the one hand, some MBs of the entire set could occasionally become unbound on the labelled DNA due to competition for the target site by a complementary genomic strand, leading to non-continuous structures in the STORM images. On the other hand, the nuclear genome has many 200 bp fragments that form nucleosomes on 10 nm 'beads on a string' chromatin fibers, which could further fold into more complex structures (30 nm fiber) (*Grigoryev and Woodcock, 2012*; *Ricci et al., 2015*). Hence, some MBs on a fragment of the target sequence might not be resolvable and would appear as distinct bright fluorescence dots, given the resolution of our 3D-STORM was 20–30 nm in lateral directions.

Similar to Oligopaint-FISH, the current method is developed from widely-used oligonucleotide-based FISH (*Yamada et al., 2011*). Its key principle is to enable probe accessibility to the target sequence, while maintaining the chromatin architecture via forming formaldehyde-mediated covalent cross-links between DNA and protein. With a long-term goal of visualizing topologic structure (connector) rather than geometrical structure (size, shape), this method puts great efforts into avoiding the disruption of the covalent cross-links to preserve the chromatin topology well, even though

there is potentially a minimal level of inter-chromatin space reduction or chromatin swelling induced by protected freeze-thawing as well as heat denaturation (*Markaki et al., 2012*; *Solovei et al., 2002*). The formaldehyde-mediated cross-link is sensitive to heat rather than low temperature (liquid nitrogen) or organic solvent (formamide or ethanol). To minimize the cross-link reversals in fixed samples and hence to maintain the nuclear ultrastructure, we perform hybridization at remarkably low temperature (22°C), compared to 42−47°C in Oligopaint-FISH (*Beliveau et al., 2015*, *2012*; *Boettiger et al., 2016*; *Wang et al., 2016*) or 37°C in other reported high-resolution FISH (*Markaki et al., 2012*; *Yamada et al., 2011*). As indicated by previous report (*Kennedy-Darling and Smith, 2014*), significantly less DNA was dissociated during 20 hr hybridization at 23°C than that at 37°C or at 47°C, indicating less disruption of formaldehyde cross-links under our hybridization condition. In addition, we also decrease temperature of washing step and remove the 0.1N HCl treatment for our MB-FISH. Under these modified conditions, chromatin topology after MB-FISH is expected to be maintained relatively well, compared to other methods like Oligopaint-FISH that was reported to provide results consistent with 3C-based technology data (*Wang et al., 2016*). Unless the cell is frozen (e.g. cryo-EM), it is so far hard to find a better way to preserve exact 3D chromatin structure (*Branco et al., 2008*). It would be great if we could combine MB-labeling with cryo-EM technique in the future.

## Materials and methods

### SK-N-SH cell culture, virus package and infection, and flow cytometry

Human SK-N-SH cell line (ATCC, HBT11) was distributed by the Cell Bank of the Chinese Academy of Sciences (Shanghai, China), and was authenticated by STR profiling and tested for the myco-plasma contamination status. The cells were cultured on coverslips (Fisherbrand Coverglass for Growth Cover Glasses 12-545-82, Fisher, Pittsburgh, PA) in MEM medium (Gibco, Waltham, MA) supplemented with 10% fetal calf serum. A lentiviral pLL3.7 vector was modified by removing the U6 promoter and puromycin resistance gene, and adding a MV promoter and downstream EGFP gene. To obtain lentiviral particles, $2 \times 10^6$ 293 T cells were co-transfected with 10 μg of the modified len-tiviral plasmid and 3.3 μg of each packaging vector (pMD2G-VSVG, pRSV-REV, and pMDL g/p RRE) using the calcium phosphate precipitation method. After 48 hr post-transfection, the supernatant from the transfectants was collected and filtered through a 0.45 μm filter (Corning, Corning, NY). The resultant virus-containing supernatant was added to the SK-N-SH cell culture and incubated for 24 hr. EGFP-positive cells were sorted by flow cytometry (BD FACSAria SORP Cell Sorter, BD Bio-sciences) using a 488 nm laser.

### PCR confirmation of lentiviral DNA integration

Blank and EGFP cells were subjected to genomic DNA extraction (PureLink Genomic DNA Mini Kit, Invitrogen, Waltham, MA) followed by confirmation of the viral DNA integration by PCR. The 30 μL PCR reaction contained 1x PrimeSTAR GXL Buffer (R050A, TaKaRa, Japan), 200 μM dNTP Mixture, 0.6 μL PrimeSTAR GXL DNA Polymerase (R050A, TaKaRa), 10 ng genomic DNA template, and 0.2 μM forward and reverse primers. Insertion checking primers were designed to target the 3.3 kb inserted viral DNA to confirm the viral integration. Control primers were designed to target the 1.7 kb fragment of the endogenous human *ACTN1* gene. The checking primers for viral DNA integration were forward, GATCTTCAGACCTGGAGGAGGAGATATG and reverse, GTCTCGATCGAGG TCGACGGTATCGATG; and the control primers for *ACTN1* gene were forward, CGGACCGA-GAAACTGCTGGAGACC and reverse, GGAACAACAAGGCGACTTTCAGGATGG.

### PCR exclusion of tandem viral repeats integration

Checking primers (PF and PR) were designed to check if there were tandem viral repeats integrated in the genome of EGFP cells. Products could be not amplified if there was only a single copy inser-tion at a genomic locus because the checking primers were outward-facing, whereas multiple prod-ucts of varying sizes could be amplified if viral concatemers were present (*Figure 1—figure supplement 2a* to c). Genomic DNA from blank or EGFP cells were subject to PCR and the results confirmed that there is no tandem viral integration at the same locus in EGFP cells (*Figure 1—figure supplement 2d*). In brief, the gel showed that no specific PCR bands were detected in EGFP cells

using the outward-facing checking primers (lane 1–3) even though a faint ~4 kb non-specific band was observed in both EGFP and blank cells (lanes 2 and 4). In addition, two additional control primers were used to ensure the checking primers were working properly by separately amplifying a 0.9 kb and 1.6 kb region of viral DNA in EGFP samples (lanes 5 and 8) but not in the controls (lanes 4 and 7). EGFP-cells confirmed without tandem repeats insertion were used in the imaging experiments. The checking primers for screening tandem viral repeats were PF, CTGCTGCCCGACAAC-CAC and PR, CGGAGTTGTTACGACATTTTGGAA. The control primer paired with PF for 0.9 kb viral DNA target was reverse, GTCTCGATCGAGGTCGACGGTATCGATG and the control primer paired with PR for 1.6 kb viral DNA target was forward, GATCTTCAGACCTGGAGGAGGAGATATG.

## ESC culture and CRISPR/Cas9-mediated knockout and verification

Mouse ESC (CJ9) was provided by Dr Xiaohua Shen (Tsinghua University, China) and cultured as previously described (*Yin et al., 2015*). Briefly, cells were maintained on 0.15% gelatin-coated (Sigma-Aldrich, St.Loius, MO) tissue plates in complete ESC culture medium containing DMEM (Dulbecco's modified Eagle's medium, Corning) supplemented with 15% heat-inactivated fetal calf serum (FCS, Foundation), 1% nucleoside mix (100x stock, EmbryoMax, Millipore, Billerica, MA), Penicillin-Streptomycin Solution (100x stock, Life Technologies), 2 mM Glutamax (100x, Life Technologies), 0.1 mM MEM non-essential amino acid (100x stock, Life Technologies, Waltham, MA), 0.1 mM 2-mercaptoethanol (Gibco), and 1000 U/mL recombinant Leukemia inhibitory factor (LIF, Millipore).

For CRISPR/Cas9-mediated knockout, two single guide RNAs (sgRNA) flanking the ~3 kb region (Chr6: 122612568–122615591, 3023 bp, mm9) of the *Nanog* locus (*Blinka et al., 2016*) were designed using the CRISPR design tool (http://crispr.mit.edu/). The sgRNA targeting sequences were separately cloned into the sgRNA expression vector ('pGL3-U6-gRNA-Puromycin mut Bsal ACCG') and subsequently co-transfected with the Cas9 expression vector ('pST1374-N-NLS-flag-linker-Cas9') into $2 \times 10^5$ ESCs by electroporation (Nucleofector technology, Lonza) (*Shen et al., 2013*). We used a ratio of 400 ng of Cas9 expression vector to 400 ng of each sgRNA plasmid. After 24 hr post-transfection, transfectants were selected with 4 µg/mL puromycin (InvivoGen, San Diego, CA) for 2 days and then plated onto 10 cm plates at clonal density. Individual clones (96 clones) were picked, expanded and subjected to genomic DNA extraction (PureLink Genomic DNA Mini Kit, Invitrogen) followed by PCR screening. A pair of inner primers designed inside of the sgRNAs pair and flanking the 2.5 kb MB-labeled region were used to confirm biallelic or monoallelic deleted clones. Absence of ~2.5 kb PCR amplification band indicated biallelic deletion, whereas presence of PCR amplification band indicated non-deleted allele. Primers amplifying ~1 kb of mouse *Actb* gene region were used as the control. After screening by genomic PCR, homozygous knockout clones were selected and sequenced for the genome region flanking the deletion, using outer primers to make sure precise target excision. The sequenced homozygous knockout clones (HoKO cells) were selected and used in the imaging experiments. The sequences of sgRNAs for endogenous CRISPR-mediated knockout and PCR primers for verification are listed as follows. Targeting sequences of sgRNAs (PAM sites are underlined) were #1, GTGTGCCGGCGCACGTGCTG<u>AGG</u> and #2, TGACATCATACAGACCGAGA<u>AGG</u>. Inner primers for PCR screening of deletion were forward, CTCCAGTCGTGGGCTAAACTGTC and reverse: GTTGACCTATAGCCAGCCACAC. Outer primers for genome sequencing of deletion regions were forward, TTACGGTCAATGATCAGAACCCATG and reverse, TTCCCCATGACATCACCCAAC. Control primer for mouse *Actb* gene were forward, AGGATGGCGTGAGGGAGAGC and reverse, ATATCGCTGCGCTGGTCGTC.

## MB probe design

Each MB probe was a 56-nt single-stranded oligonucleotide containing a 42-nt target hybridizing region flanked by 7-nt non-genomic arms that were complementary to each other (*Figure 1a*, top). Hybridizing regions in the different MBs were designed to be complementary to specific sites within the target sequence (*Rouillard et al., 2003*). The conditions were as follows: (i) a melting temperature above 70℃, (ii) no more than 25 nt complementarity to genomic sequences elsewhere using BLAST+, (iii) no contiguous repeats of six or more identical nucleotides, and (iv) no secondary structure formation at the hybridization temperature or higher. The two flanking arms were designed to form a stable stem structure at the hybridization temperature in the absence of the target site. General considerations for the arm sequence were as follows: (i) the two arms were GC rich (85%) and

complementary to each other and (ii) instead of G, a C was used at the 5' end next to the Alexa-647 fluorophore, because G was reported to have a quenching effect on the fluorophore (*Kelley and Barton, 1999*; *Nazarenko et al., 2002*; *Seidel et al., 1996*).

The folding properties of the entire MB sequence were predicted by the UNAfold package (*Markham and Zuker, 2008*) to check whether the intended hybridizing region/arm conformation could function under the hybridization conditions. The specific requirements for the MB design were as follows: (i) the predicted melting temperature of arm-arm binding should be within a narrow melting range between 50°C and 60°C, (ii) secondary structure between the hybridizing region should be avoided as it might interfere with hairpin formation and arm sequences were changed if there was unwanted secondary structure, and (iii) no formation of stems longer than intended, which might slow binding of MB to the target. For designing the specific MBs for tiling along the target, we used BLAST+ and the UNAfold package (*Markham and Zuker, 2008*) and excluded probes that were partially complementary to each other, which might cause unintended binding.

## Spectrophotometric measurement of MB fluorescence intensity

Individual MBs and their corresponding complementary sequence (CS) (Life Technologies) were dissolved in buffer containing 50 mM NaCl, 1 mM EDTA, 10 mM Tris (pH 7.4) and made to a 10 μM stock. For each MB, we mixed CSs of the other 28 MBs (10 μM stock) as the non-complementary sequences (NCSs), which shared $10 \pm 5$ nt complementarity with the MB. Triplicate samples of each MB (80 nM) were incubated in FISH hybridization buffer containing 2x SSC and 50% formamide in the presence of corresponding CS (1600 nM) or NCSs (1600 nM) at the indicated temperature for 30 min. Alexa-647 fluorescence intensity of each MB reaction was measured three times with excitation at 647 nm and emission at 665 nm (Varioskan Flash 4.00.53).

## Preparation of cell samples with relatively uniform distribution of Alexa-647

Human SK-N-SH cells were grown on coverslips to around 80% confluence and fixed with ice-cold 4% formaldehyde/PBS (freshly prepared from paraformaldehyde) for 10 min, and then treated with 1 mg/mL sodium borohydride for 7 min, followed by 0.5% Triton-X-100 in PBS for 10 min. Each coverslip was then incubated at 4°C with 65 μL ddH2O containing 5'-TTTTTTTTTT-3' (10T) or Alexa-647-conjugated 10T (a 10T oligonucleotide labeled with one Alexa-647 at its 5' end) at the indicated concentration to allow for non-specific association of oligonucleotide 10T within the nuclear genome in the cells. Samples with sparse or extremely sparse Alexa-647 density were incubated overnight with 200 pM or 80 pM of Alexa-647-conjugated 10T, respectively. Samples with relatively high Alexa-647 density were incubated with 2 nM of Alexa-647-conjugated 10T for 2 hr. Samples were rinsed with 2x SSC to remove free dye, and then fixed with 4% formaldehyde/PBS for 10 min and kept at 4°C for imaging.

## 3D-STORM imaging

A STORM system based on an inverted optical microscope (IX-71, Olympus) with a 100× oil immersion objective lens (UplanSApo, N.A. 1.40, Olympus) was used for the nanoimaging as previously described (*Huang et al., 2008b*). Astigmatism imaging method was adopted for 3D-STORM, in which a weak cylindrical lens (1 m focal length) was introduced into the imaging path. A 405 nm laser (CUBE 405–100C; Coherent) was used for photoactivation and a 641 nm laser (CUBE 640–100C; Coherent) was used to excite fluorescence and switch Alexa-647 to the dark state. The illumination used the highly inclined and laminated optical sheet (HILO) configuration (*Tokunaga et al., 2008*). The laser power densities were approximately 18 W/cm$^2$ for the 405 nm laser and 1 kW/cm$^2$ for the 641 nm laser unless otherwise indicated. A dichroic mirror (ZT647rdc, Chroma) was used to separate the fluorescence from the laser and a band-pass filter (FF01-676/37, Semrock) on the imaging path was used to filter the fluorescence. Raw images of the fluorescent signals in each nuclear field were acquired with an EMCCD (DU-897U-CV0, Andor) at 85 Hz for 36000 frames (two series of 18000 frames) unless otherwise indicated. The 641 nm laser was used during the whole imaging process, and the 405 nm laser was added in the second series of images. To avoid focal drift, an anti-drift system was used to sustain the focal position within 10 nm during image processing (*Huo et al., 2015*). Unless specified, a standard STORM imaging buffer containing 50 mM Tris (pH 8.0), 10 mM NaCl,

1% $\beta$-mercaptoethanol (v/v), 10% glucose (w/v), 0.5 mg/mL glucose oxidase (G2133, Sigma), and 40 µg/mL catalase (C30, Sigma) (*Dempsey et al., 2011*; *Olivier et al., 2013*) was used. A modified imaging buffer containing 50 mM Tris (pH 7.5), 10 mM NaCl, 1% $\beta$-mercaptoethanol (v/v), 10% glucose (w/v), 0.25 mg/mL glucose oxidase (G2133, Sigma), and 20 µg/mL catalase (C30, Sigma) was also used (*Figure 2c*).

## STORM image reconstruction and identification

The STORM imaging data were analyzed in a similar manner as previously described (*Huang et al., 2008b*). Briefly, the fluorescence images were preprocessed using a band-pass filter to reduce the background noise, and then fitted to an elliptical Gaussian function. The position of the fluorophore in the x-y plane was directly read out by the function, while the axial position was calculated from an experimental calibration curve of ellipticity versus z position. Next, a cleaning procedure was performed to remove unqualified fluorescence spots by adjusting the thresholds for photons (<300), PSF size (<4 pixels), correlation index (<0.95), and simultaneous localization density (<1.2 µm). The qualified events were recorded and their corresponding coordinates in 3D were determined. To correct for sample drift in the x-y plane, 0.2 µm diameter red fluorescent microspheres (F8810, Thermo Fisher) were added to the samples as fiducial markers during sample imaging. The positions of multiple fluorescent beads in each view field were calculated by the same fitting algorithm. A drift-corrected image was obtained from the drift trace by subtracting the averaged motion of the beads. Each super-resolution image was reconstructed from one or two series of sequential frames. In the reconstructed image, each localization was represented as a Gaussian peak, whose standard deviation was matched with the localization precision and intensity was normalized to photons. The z-position information was presented as pseudo-colors according to the color bar. Only the regions containing 100–3000 localizations were reconstructed into the super-resolution images and subjected to further filtering analysis and verification. A single Alexa-647 fluorophore gives rise to 10–100 localizations during the STORM acquisition time under our experimental conditions (data not shown).

## MB-FISH

Human SK-N-SH cells or mouse ESCs were cultured on poly-lysine-coated coverslips for 16–20 hr to around 80% confluence, and then fixed with 4% freshly prepared cold formaldehyde/PBS for 10 min, followed by soaking in PBS for 2 min (*Hoffman et al., 2015*). Cells were treated with 1 mg/mL sodium borohydride in ddH2O for 7 min, followed by soaking in ddH2O for 2 min. Cells were immersed in 25% glycerol-PBS for 40–50 min, and then frozen in liquid nitrogen and thawed in air, and this freeze-thaw cycle was repeated three times (*Markaki et al., 2013*, *2012*; *Solovei et al., 2002*). Cells were incubated with Rnase A (100 µg/mL) for 1 hr at 37°C and then rinsed with PBS. Cells were soaked in PBS for 5 min and then pre-warmed at 75°C for 5 min in 2x SSC buffer replaced with 80% deionized formamide in 2x SSC buffer for a further 3 min. Cells were then immediately incubated in a cold ethanol series (75–90–100%) at 2 min per step (*Hogan et al., 2015*). Cells were blocked with hybridization buffer containing 50% formamide, 1.5% FISH blocking powder (Roche, Switzerland) and 2x SSC overnight at room temperature (22°C). At 3 to 6 hr before hybridization, the hybridization buffer was replaced with the same hybridization buffer at 42°C. Subsequently, each sample on a 12 mm coverslip was mounted in 14 µL hybridization buffer containing 1 µL MB mixture (10 µM) on a slide using rubber cement (Elmer's, 72170). The hybridization reaction was allowed to occur for 16–20 hr. Although the optimal temperature range was 22°C to 14°C as suggested by the in vitro MB fluorescence measurements, the higher temperature was able to increase the MB probe availability to the genomic target sequence. The hybridization temperature of 22°C was chosen as this was the room temperature of our laboratory and for convenience in the following washing step. Cells were washed with buffer containing 50% formamide and 2x SSC for 40–50 min at room temperature and fixed in 4% formaldehyde/PBS for 5–10 min. Cell were then kept in 0.25x SSC buffer and stored in a fridge until the STORM imaging. The SSC buffer was diluted from a 10x SSC stock (1 L: 87.69 g NaCl and 50.2 g sodium citrate dihydrate).

## Statistical analysis

Statistical analyses comparing Alexa-647 detection efficiency among imaging conditions (*Figure 2*) were conducted using two-sided two-sample t tests. Data were expressed as mean ± SEM as indicated. A pvalue<0.05 was considered statistically significant. All experiments involving statistical analysis were performed with three biological replicates. Statistical significance was labeled as follows: *$p<0.05$, **$p<0.01$, ***$p<0.001$. The specificity of MB-FISH in positive and negative control samples was assessed using the hypergeometric p value.

## Acknowledgements

We thank XH Shen and YF Yin for providing plasmids and advice on culturing ES cells; CH Yu, JT Gao, KK Tsia, HK So, J Man and JL Qu for their constructive discussion; H Zhang, JJ Hu and M Wang for their support with the image presentation; and C Xu, H Li, YC Cai, YY Zhao, and N Zhou for their technical assistance. This work was supported by National Basic Research Program of China (2012CB825802), the National Natural Science Foundation of China (61235012, 31401146, 31361163004, 91019016,31671384), the Special-Funded Program on National Key Scientific Instruments and Equipment Development (2012YQ150092), the National Basic Research Program of China (2012CB316503, 2015CB352005), the National Natural Science Foundation of China (61178080, 61335001), and the Shenzhen Science and Technology Planning Project (JCYJ20150324141711698).

## Additional information

### Funding

| Funder | Grant reference number | Author |
| --- | --- | --- |
| National Basic Research Program of China | 2012CB825802 | Hanben Niu |
| National Natural Science Foundation of China | 61235012 | Hanben Niu |
| National Natural Science Foundation of China | 31401146 | Yanxiang Ni |
| National Natural Science Foundation of China | 31361163004 | Michael Q Zhang |
| National Natural Science Foundation of China | 91019016 | Michael Q Zhang |
| National Natural Science Foundation of China | 31671384 | Michael Q Zhang |
| Special-Funded Program on National Key Scientific Instruments and Equipment Development | 2012YQ150092 | Hanben Niu |
| National Basic Research Program of China | 2012CB316503 | Michael Q Zhang |
| National Basic Research Program of China | 2015CB352005 | Danni Chen |
| National Natural Science Foundation of China | 61178080 | Bin Yu |
| National Natural Science Foundation of China | 61335001 | Danni Chen |
| Shenzhen Science and Technology Planning Project | JCYJ20150324141711698 | Bin Yu |

The funders had no role in study design, data collection and interpretation, or the decision to submit the work for publication.

## Author contributions
YN, Conceptualization, Data curation, Formal analysis, Validation, Investigation, Visualization, Methodology, Writing—original draft, Project administration; BC, Data curation, Software, Formal analysis, Validation, Investigation, Visualization, Methodology, Writing—original draft; TM, Data curation, Formal analysis, Validation, Investigation, Visualization, Methodology, Writing—original draft; GN, Data curation, Software, Formal analysis, Validation, Investigation, Methodology, Visualization; YH, Software, Formal analysis, Methodology; JH, DC, Writing—review and editing; YL, Writing—review and editing, Methodology ; BY, Investigation; MQZ, Data curation, Resources, Supervision, Funding acquisition, Methodology, Writing—review and editing, Formal analysis; HN, Data curation, Resources, Supervision, Funding acquisition, Methodology, Software, Formal analysis

## Author ORCIDs
Michael Q Zhang, http://orcid.org/0000-0002-7408-1830

## Additional files

### Major datasets
The following dataset was generated:

| Author(s) | Year | Dataset title | Dataset URL | Database, license, and accessibility information |
|---|---|---|---|---|
| Yanxiang Ni, Bo Cao, Tszshan Ma, Gang Niu, Yingdong Huo, Jiandong Huang, Danni Chen, Yi Liu, Yu Bin, Michael Q Zhang, Hanben Niu | 2017 | Data from: Super-resolution imaging of a 2.5 kb non-repetitive DNA in situ in the nuclear genome using molecular beacon probes | http://dx.doi.org/10.5061/dryad.126k9 | Available at Dryad Digital Repository under a CC0 Public Domain Dedication |

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
