## [Decision Letter]

[Editors’ note: this article was originally rejected after discussions between the reviewers, but the authors were invited to resubmit after an appeal against the decision.]

Thank you for submitting your work entitled "Super-resolution imaging of a 2.5-kb non-repetitive DNA *in situ* in human genome using molecular beacon probes" for consideration by *eLife*. Your article has been reviewed by four peer reviewers, and the evaluation has been overseen by a Reviewing Editor and a Senior Editor. The following individuals involved in review of your submission have agreed to reveal their identity: Lothar Schermelleh (Reviewer #1).

Our decision has been reached after consultation between the reviewers. Based on these discussions and the individual reviews below, we regret to inform you that your work will not be considered further for publication in *eLife*.

While all reviewer found the presented method of potential value for the community, they also agree that detection of an endogenous locus is an absolute requirement for showing the utility and the specificity of the method. There was also a common concern about the lack of detail on the efficiency of the approach and the protocol itself.

Whilst we are rejecting the paper for now due to the amount of work that needs done, we would be happy to receive a revised paper as a new submission once all the reviewers' concerns have been addressed. Please note that in this case *eLife* cannot guarantee that a resubmitted paper will be accepted and that it is the authors' choice whether to submit their manuscript to another journal instead of performing the additional work.

Reviewer #1:

Ni et al. present a novel "molecular beacon" fluorescence *in situ* hybridization (MB-FISH) method, that reduces genomic target size down to 2.5 kb, about half of the previously best DNA-FISH approaches. They achieve this by tailoring synthetic oligonucleotide probes with a quencher – fluorophore pair at opposite ends flanked by short complementary sequences. By this smart trick, probes only fluoresce upon hybridisation with the target sequence, while unbound probes remain largely non-fluorescent, thus reducing unspecific background and improving signal to noise.

As the method was developed with super-resolution 3D-STORM imaging in mind, the authors also devise an improved protocol for STORM imaging of sparse target labeling, which is a very useful add-on to this paper.

The presented MB-FISH method will without doubt generate considerable interest in the field. The experimental logic, technical execution, data quality and conclusions are largely sound. The manuscript itself is clearly presented and written concisely. However, I feel that the following issues need to be addressed first to firmly justify publication in *eLife*'s Tools & Resources section.

Major comments:

As proof-of-principle the authors use their MB-FISH approach to detect a randomly integrated transgenic sequence encoding for eGFP linked to a strong constitutive CMV promoter. While this is a very suitable approach to have a clean negative control to compare with, it does not show how reliable the method is in detecting true endogenous sequences. Furthermore, CMV-driven constitutive expression will likely render the underlying chromatin highly decondensed and accessible. What is the ability of MB-FISH to detect non-expressed sequences of the same size?

Hence, in order to convey full confidence in this new method, the authors need to demonstrate the detection of one, or preferably a few, endogenous target sequences, ideally representative for different classes (genic, intergenic, expressed, non-expressed).

Reviewer #2:

This paper describes the application of molecular beacon probes to super-resolution imaging by STORM. The authors optimised hybridisation temperature in solution using complementary sequence. They optimised imaging conditions by exposing fixed and permeabilised cells to Alexa 647-tagged 10nt oligos followed by wash and fix to leave sparse Alexa 647 signals for imaging and establishment of the resolution achieved. Finally they present example STORM images of the transfected viral sequences.

This is a straightforward technical paper describing the smallest unique sequence yet detected by super-resolution imaging, and as such the method deserves to be published. I do have suggestions however that will hopefully improve the manuscript. I do not have the expertise to properly assess the home-built STORM system and image reconstruction – hopefully another reviewer will address this.

Major Comments

The target DNA of 2.5kb is not endogenous human but viral CMV plus eGFP. This will minimize the background. It would be helpful to include detection of a human sequence to indicate sensitivity within a more likely experimental situation.

Similarly, I was surprised that the imaging optimisation was done on the 'sparse Alexa 647' preps rather than hybridized signal and it would be helpful to have some rationale here. What was the 10nt sequence used and why? Does each cluster imaged represent a single fluorophore? The images should be presented.

I would like to see more examples of the MB signals (I could not access the repository) and more discussion of how true signals were distinguished from unbound/off target signal. For example, comparing the yellow signals in Figure 4 and Figure 5, 3rd row – how were they distinguished as non-specific or specific signal?

Controls other than the blank cells should be included e.g. scrambled oligos.

I felt that the Discussion lacked depth. There was disappointingly no discussion of the nanostructures imaged. Are any of these cells in G2? Some more discussion of appropriate fluors and schemes for multi-colour analysis would also be appreciated.

The paper does need to be overhauled by a native English speaker. There are several points of confusion, for example at the end of the Discussion on reference to Hogan et al., 2015. Do they mean that Hogan used molecular beacons or that this would be a suitable application?

Finally (and this is an observation directed not only at this paper), how can the authors be certain that nanostructures imaged after cycles of freeze-thaw will accurately represent three-dimensional chromatin conformation in the nucleus?

Reviewer #3:

Chromatin topology within the nucleus plays an important role in many biological processes such as enhancer-promoter contacts during gene expression. In this manuscript, the authors developed a super-resolution imaging method to visualize a 2.5-kb non-repetitive DNA *in situ* in human cells using molecular beacon (MB) probes. They first analyzed the efficiency of 29 MB probes targeting a 2.5-kb exogenous DNA fragment by comparing the probe fluorescence with excessive amount of complementary sequences or non-complementary sequences under different conditions. Second, they optimized conventional STORM imaging conditions to identify signals of sparse Alexa-647 from the background. Finally, they reconstructed nano-structures of the 2.5-kb non-repetitive DNA *in situ* in the human genome. To be useful for investigating chromatin looping contacts, at least two DNA elements need to be visualized at the same time. Nevertheless, this is a significant study, if true, may represent the shortest DNA fragment visualized to date in the human genome.

My major concern is that whether the method could truly detect nano-structures of the 2.5-kb endogenous non-repetitive DNA *in situ* in the human genome. Because the viral DNA could be randomly integrated into the human genome, it is not clear how the authors could use PCR to detect single copy insertions. In theory, DNA could be integrated in multiple tandem copies in random loci, there is not enough information how the authors designed PCR primers. A PCR band of around 3.3 kb rather than folds of the size is not strong evidence for single-copy insertion.

Major points:

1) Whether the non-complementary sequences could mimic the true off-targets in MB binding? It is difficult to design non-complementary sequences sharing 5-15 nucleotides with individual MBs to mimic many possible off-target MB bindings.

2) Whether the frame rate of 85 Hz and power of 0.5 mW 405-nm laser and 29 mW 641-nm laser are the best optimized condition since the values of blank cells are very low for calculating FDR? In addition, in Figure 2, 0.04 ± 0.02 basically means there is no auto-fluorescence.

3) It would be useful if the authors could show that the method could be used to study an enhancer-promoter or promoter-promoter looping contact as a proof-of-principle.

Reviewer #4:

In their manuscript, Zhang et al. develop a FISH approach based on molecular beacons that can be used to label short (2.5kb) non-repetitive regions of genomic loci for 3D super-resolution imaging. The approach is interesting and can provide a simpler method of labeling non-repetitive sequences compared to Oligopaint approach. However, I have major concerns about the specificity of the probes and the validity of the method (see comments below):

Major comments:

1) The major weakness of the manuscript is the use of cells with random integration instead of a more controlled system. This choice makes it difficult to truly assess the specificity of the approach. The authors should have used a system of site specific integration (for example: CRISPR-Cas9, FlpIn recombination, Integrations in R26 locus). Alternatively, the authors could have targeted an endogenous region and use as control the knockout counterpart. Wild type vs KO cell lines are available for a variety of genes or can be generated with CRISPR-Cas9. In these systems the authors should detect a number of loci compatible with the ploidy of the cell. In addition, in such a system the authors can confirm the specificity of the molecular beacons by labeling the same genomic locus with an alternative method such as regular FISH. In the absence of these experiments, the specificity of the molecular beacons remains unconfirmed and not convincing from the provided data.

2) The manuscript is also largely descriptive and lacking quantitative information to back up the major claims. For example, was the viral MOI (multiplicity of infection) assessed? From this value at least a range of expected integration sites could be estimated and compared to the super-resolution data.

The authors should also provide more information regarding the efficiency of their approach:

– what is the percentage of GFP positive cells with detectable loci?

– what is the average number of detected loci? Is this number in line with the viral MOI used?

– what is the average and range of detected localisations per locus?

In general, representative images should be supported with more quantitative data based on the complete dataset of cells imaged. It would also be useful to provide unzoomed images to appreciate the general signal to noise ratio in the whole ROI.

3) When comparing MB-FISH to Oligopaints, the authors should further discuss the differences/similarities between the two approaches:

– can MB-FISH provide allele specificity and is it sensitive to SNPs.?

– what is the optimal probe density required for MB-FISH?

– are there specific requirements in terms of spacing between probes, strand specific orientation, etc?

– would this approach be compatible with live imaging?

4) The usage of English language in the manuscript needs substantial improvement. The grammatical errors and awkward sentences are far too many to point out one by one in this review.

[Editors’ note: what now follows is the decision letter after the authors submitted for further consideration.]

Thank you for resubmitting your work entitled "Super-resolution imaging of a 2.5 kb non-repetitive DNA *in situ* in the nuclear genome using molecular beacon probes" for further consideration at *eLife*. Your revised article has been favorably evaluated by Jessica Tyler (Senior editor), a Reviewing editor, and two reviewers.

The manuscript has been improved but there are some remaining issues that need to be addressed before acceptance, as outlined below:

Reviewer #1:

The authors addressed most issues raised in my first review. However to meet the highest standards of *eLife*, a few minor issues still need to be resolved:

1) Sections of the manuscript have become somewhat cluttered and difficult to digest, in part by addressing the reviewers' additional requests. For instance, experimental details in the Results section (e.g. cloning controls) could potentially be moved to the Methods section. Also the Discussion could be streamlined (see also point 3). Generally I feel the manuscript would benefit from another round of decluttering/shortening to present this excellent work in the most concise and clearest way.

2) I am still not entirely clear how to interpret the success rate of 21/78 and 14/92 cells with detected nanostructures. Does this reflect the limitation of the STORM imaging method to be able of imaging only a sub-volume of a mammalian nucleus? Or does is reflect (also) a lowered detection efficiency of the FISH method. Or is it a combination of both? What is the chance of detecting positive cells, if one assumes one, two (or more) loci, if the detection would be 100% efficient? The authors may still need to clarify this better.

3) The use of language needs improvement (particular in the revised Discussion).

Reviewer #2:

The majority of my questions have been answered, in particular the authors have analysed an endogenous locus. However there remain outstanding issues that require explanation and/or addressing in the text for me to feel confident of the validity of the signals presented. Criteria for image analysis and some of the data produced do not appear to be presented fully.

1) Response to Reviewer 2 Point 6: In fact the Markaki 2012 paper clearly states (page 415, last paragraph): "In agreement with our visual impression, the IC space was reduced in 3D-FISH-treated nuclei and a shift toward higher intensity classes. […] We also consider the possibility that some swelling or dispersal of chromatin, in particular as a result of the heat denaturation step [18], resulted in an improved accessibility of DNA to this fluorophore." Please could the authors clarify this point in the Discussion.

2) Results and Figure 4 wondered why the beads (F8810 580/605) were visible at 405nm and occasionally at 641nm?

3) Results subsection “Super-resolution visualization of 2.5 kb enhancer *in situ* in Nanog locus of mouse ESCs” and Figure 5—figure supplement 1: It is not clear what point the authors are making about the nuclear periphery sitting close to the cell surface. Please clarify.

4) Figure 4: What is the blue dot in Cell I E?

5) Figure 5 Panel C Cell V: The position of the nanostructure in the box in C at low resolution does not appear to be the same as in D. Please explain.

6) Results: The authors provide statistical data on larger series than the 14/92 ESCs and the 21/78EGFP cells. Why are the numbers in the larger series not provided?

7) Discussion paragraph three: Please give the limits in localisation number and area that were used as criteria for defining nanostructures.

8) Discussion paragraph five: This explanation could perhaps be clarified.

9) Discussion paragraph six: The authors could perhaps refer to Ricci et al. here (PMID: 25768910).

Reviewer #3:

In the revision, the authors have performed additional experiments to show the proof-of-principle usage of their MB-FISH method. First, they have visualized a 2.5 kb non-repetitive endogenous DNA *in situ* in the human genome. Second, they have tested the method on a 2.5 kb super-enhancer at -45 kb upstream within the Nanog locus in mouse embryonic stem cells. As super-enhancers are clusters of enhancers which are located physically close in a 3D genome, the MB-FISH may be useful to study gene expression in the future. Finally, they applied CRISPR/Cas9-mediated knockout in mESCs to generate the homozygous knockout (HoKO) cells for the negative control experiments of the super-enhancer MB-FISH, in which a 3 kb region covering the super-enhancer target is deleted from both mouse alleles. Thus, the authors have largely addressed my concerns.

---

## [Author Response]

[Editors’ note: the author responses to the first round of peer review follow.]

*Reviewer #1:*

*Major comments:*

*As proof-of-principle the authors use their MB-FISH approach to detect a randomly integrated transgenic sequence encoding for eGFP linked to a strong constitutive CMV promoter. While this is a very suitable approach to have a clean negative control to compare with, it does not show how reliable the method is in detecting true endogenous sequences. Furthermore, CMV-driven constitutive expression will likely render the underlying chromatin highly decondensed and accessible. What is the ability of MB-FISH to detect non-expressed sequences of the same size?*

*Hence, in order to convey full confidence in this new method, the authors need to demonstrate the detection of one, or preferably a few, endogenous target sequences, ideally representative for different classes (genic, intergenic, expressed, non-expressed).*

We appreciate the reviewer for the important comments and suggestions, which are very helpful for our revision work. As a response, we chose a 2.5-kb non-coding genomic region (a super-enhancer within *Nanog* locus) (Blinka et al., 2016) in the mouse genome as our new target, designed a set of 34 MBs according to the target sequence, and applied MB-based super-resolution FISH (MB-FISH) to visualize the endogenous target in mouse embryonic stem cell (ESC). Furthermore, we also applied CRISPR/Cas9-mediated knockout to generate a homozygous knockout (HoKO) negative control cell, in which a ~3 kb DNA fragment covering the 2.5- kb imaged target was deleted from both *Nanog* alleles. In each cell, we collected and analysed STORM imaging data from a layer (700 nm in depth) of the nucleus. Based on the previous version, we have now applied stringent strategies in obtaining nanostructures. For example, conventional imaging of beads is now provided to confirm that no identified nanostructure is overlapping with or close to fluorescent bead used for drift correction. After image reconstruction and candidate filtering, we detected 14 cells with MB-labelled nanostructure(s) from 92 WT ESCs but zero from 74 HoKO controls. When further quantifying the MB-FISH specificity by random sampling using the same criteria with larger samples, we found the p-value of 1.49e-04 for this endogenous experiment. In the future studies, more MB-labelled target sequence will be identified by collecting and analyzing complete datasets from the whole nucleus (at least 5 µm in depth) rather than only from a single nuclear layer (700 nm in depth) of each nucleus. On the other hand, although we had carried out STORM imaging in dozens of cells in the last submission, we have recently performed more in EGFP or blank SK-N-SH cells. During the revision, we identified 7 from 28 EGFP cells to be with detected nanostructure, one nanostructure in each positive cell; whereas zero from 33 blank controls. Collectively, we have so far identified 21 or zero cells to be with detected nanostructure from 78 EGFP or 83 blank cells, respectively, giving a p value of 1.00e-6 when random sampling using the same criteria with larger samples. Taken together, these data show the capability and robustness of our MB-FISH in visualizing a unique genomic sequence in the human or mouse genome. For the details, please see last two sections of Result and last three paragraphs of Discussion in the current version.

*Reviewer #2:*

*The target DNA of 2.5kb is not endogenous human but viral CMV plus eGFP. This will minimize the background. It would be helpful to include detection of a human sequence to indicate sensitivity within a more likely experimental situation.*

We thank the reviewer for the critical comments, which help us to substantially improve our work. As a response, we chose a 2.5-kb non-coding genomic region (a super-enhancer within *Nanog* locus) (Blinka et al., 2016) in the mouse genome as our new target, designed a set of 34 MBs according to the target sequence, and applied MB-based super-resolution FISH (MB-FISH) to visualize the endogenous target in mouse embryonic stem cell (ESC). Furthermore, we also applied CRISPR/Cas9-mediated knockout to generate a homozygous knockout (HoKO) negative control cell, in which a ~3 kb DNA fragment covering the 2.5- kb imaged target was deleted from both *Nanog* alleles. In each cell, we collected and analysed STORM imaging data from a layer (700 nm in depth) of the nucleus. Based on the previous version, we have now applied stringent strategies in obtaining nanostructures. For example, conventional imaging of beads is now provided to confirm that no identified nanostructure is overlapping with or close to fluorescent bead used for drift correction. After image reconstruction and candidate filtering, we detected 14 cells with MB-labelled nanostructure(s) from 92 WT ESCs but zero from 74 HoKO controls. When further quantifying the MB-FISH specificity by random sampling using the same criteria with larger samples, we found the p-value of 1.49e-04 for this endogenous experiment. In the future studies, more MB-labelled target sequence will be identified by collecting and analyzing complete datasets from the whole nucleus (at least 5 µm in depth) rather than only from a single nuclear layer (700 nm in depth) of each nucleus.

On the other hand, although we had carried out STORM imaging in dozens of cells in the last submission, we have recently performed more in EGFP or blank SK-N-SH cells. During the revision, we identified 7 from 28 EGFP cells to be with detected nanostructure, one nanostructure in each positive cell; whereas zero from 33 blank controls. Collectively, we have so far identified 21 or zero cells to be with detected nanostructure from 78 EGFP or 83 blank cells, respectively, giving a p value of 1.00e-6 when random sampling using the same criteria with larger samples.

Taken together, these data show the capability and robustness of our MB-FISH in visualizing a unique genomic sequence in the human or mouse genome. For the details, please see last two sections of Result and last three paragraphs of Discussion in the current version.

*Similarly, I was surprised that the imaging optimisation was done on the 'sparse Alexa 647' preps rather than hybridized signal and it would be helpful to have some rationale here. What was the 10nt sequence used and why? Does each cluster imaged represent a single fluorophore? The images should be presented.*

We thank the reviewer for raising the concern and suggestion. To compare multiple imaging conditions and identify the optimal one, we need cell samples with uniformly distributed dyes in the entire nucleus. We did not choose MB-hybridized samples, in which signals existed only in some local compartments of nucleus. Instead, we incubated SK-N-SH cells with Alexa-647-conjugated oligonucleotide, in which the 10T oligonucleotide was used to mediate the non-specific link of Alexa-647 to the nuclear genome at 4^o^C during sample preparation. In this regard, the oligonucleotide could be any sequence except those appeared as tandem repeats in nuclear genome. We used 10T to mediate the non-specific link and found it work well, enabling Alexa-647 molecules to be distributed in a sparse and relatively uniform manner in the entire nucleus. Representative STORM images of sparse Alexa-647 distribution were now included in revised Figure 2, in which the green dot shows the fitted position of each event. Each cluster of green dots indicates multiple events collected during the whole imaging process, probably contributed by a single dye. Relative description was also included in revised Result section and Materials and method section (“Preparation of cell samples with Alexa-647 at different densities”).

*I would like to see more examples of the MB signals (I could not access the repository) and more discussion of how true signals were distinguished from unbound/off target signal. For example, comparing the yellow signals in Figure 4 and Figure 5, 3rd row – how were they distinguished as non-specific or specific signal?*

Thank the reviewer for the constructive suggestions. In addition to the images showed in the previous submission, now we included new images from our newly performed exogenous and endogenous experiments in the revised Figure 4 and Figure 5. In these new data, convention imaging of beads in the same field was also provided to ensure no nanostructure was overlapping with fluorescent bead. Furthermore, we had an intense discussion about the filtering criteria, according to which true signals were distinguished from unbound/off-target signals. As an example, signals in previous or current Figure 4 were sparsely distributed fluorophores, which gave rise to localizations only across a short series of continual frames, and were filtered out as unbound/off-target individual MB; meanwhile, nanostructure in third row of previous Figure 5 or Cell I of current Figure 4 were composed of multiple contiguous dots with localizations stochastically distributed along the whole acquisition time. For more details, please see the third and fourth paragraph of revised Discussion.

*I felt that the Discussion lacked depth. There was disappointingly no discussion of the nanostructures imaged. Are any of these cells in G2? Some more discussion of appropriate fluors and schemes for multi-colour analysis would also be appreciated.*

Thank the reviewer for the comments and suggestions. As a response, we have an intense discussion on the nanostructure imaged in last three paragraphs of revised Discussion. In regards to suitable fluorophores for MB probes, any reported dye used in STORM imaging with an available corresponding quencher could be applicable, such as Atto-488, Atto-565, Alexa-647, or Cy5 (Dempsey et al., 2011, Heilemann et al., 2009, Heilemann et al., 2008, Lehmann et al., 2015, Rust et al., 2006). In the future, it might be possible to label and image three genomic elements using sets of MBs carrying Atto-488, Atto-565, or Alexa-647, respectively, and multi-color STORM to enable detection of interactions among loci *in situ* in single cells. This could be an important approach to shed a new light in studying looping interactions and 3D chromatin organizations at super-resolution in single cells. These lines of discussion are now included into the first paragraph of the revised Discussion.

*The paper does need to be overhauled by a native English speaker. There are several points of confusion, for example at the end of the Discussion on reference to Hogan et al., 2015. Do they mean that Hogan used molecular beacons or that this would be a suitable application?*

We apologize for the inappropriate expression in the manuscript as well as for the inconvenience caused, and thank the reviewer for the patience and suggestion. We have seriously revised the manuscript and hoped that the current version could be better understood. In addition, the sentence mentioned by the reviewer is now present as “With the use of MB-FISH, it may be possible to directly visualize transient pairing of narrowed elements within Oct4 enhancer/promoter region during the onset of ESC differentiation (Hogan et al., 2015).”

*Finally (and this is an observation directed not only at this paper), how can the authors be certain that nanostructures imaged after cycles of freeze-thaw will accurately represent three-dimensional chromatin conformation in the nucleus?*

We thank the review for raising the concern. FISH is a widely-used method important for visualizing and mapping genomic locus in nucleus. It key steps are to preserve chromatin architecture while ensuring sufficient probe accessibility to the genomic target. Many experimental procedures have been developed to address the issues in the past decades. The procedure of multiple freeze-thaw cycles with the protection of glycerol is one of the approached used to facilitate probe accessibility after preserving nuclear genomic architectures with PFA fixation (Hoffman et al., 2015, Markaki et al., 2012, Boettiger et al., 2016, Markaki et al., 2013). This approach has been confirmed by electron microscopy or super-resolution imaging studies to have minimal disturbances to the chromatin ultrastructure with regard to domain size, shape, and topography (Markaki et al., 2012, Markaki et al., 2013, Solovei et al., 2002). Moreover, a recent FISH/STORM report using freeze-thaw step to study the chromatin domain spatial organizations *in situ* has shown that the imaging results agree with Hi-C studies in terms of inter-TAD distances, supporting the notion that FISH procedure still largely preserved the chromatin ultrastructure (Wang et al., 2016). Although freeze-thaw cycle was involved in our MB-FISH, we removed step of HCl treatment and decreased the temperature of hybridization and washing to maintain the original genome structure. The detailed MB-FISH protocol has been described in Methods sectionfor your information.

*Reviewer #3:*

*Chromatin topology within the nucleus plays an important role in many biological processes such as enhancer-promoter contacts during gene expression. In this manuscript, the authors developed a super-resolution imaging method to visualize a 2.5-kb non-repetitive DNA in situ in human cells using molecular beacon (MB) probes. They first analyzed the efficiency of 29 MB probes targeting a 2.5-kb exogenous DNA fragment by comparing the probe fluorescence with excessive amount of complementary sequences or non-complementary sequences under different conditions. Second, they optimized conventional STORM imaging conditions to identify signals of sparse Alexa-647 from the background. Finally, they reconstructed nano-structures of the 2.5-kb non-repetitive DNA in situ in the human genome. To be useful for investigating chromatin looping contacts, at least two DNA elements need to be visualized at the same time. Nevertheless, this is a significant study, if true, may represent the shortest DNA fragment visualized to date in the human genome.*

*My major concern is that whether the method could truly detect nano-structures of the 2.5-kb endogenous non-repetitive DNA in situ in the human genome. Because the viral DNA could be randomly integrated into the human genome, it is not clear how the authors could use PCR to detect single copy insertions. In theory, DNA could be integrated in multiple tandem copies in random loci, there is not enough information how the authors designed PCR primers. A PCR band of around 3.3 kb rather than folds of the size is not strong evidence for single-copy insertion.*

We thank the reviewer for raising the concerns, in response to which we have performed substantial work and remarkably improved our work.

First, we chose a 2.5-kb non-coding genomic region (a super-enhancer within *Nanog* locus) (Blinka et al., 2016) in the mouse genome as our new target, designed a set of 34 MBs according to the target sequence, and applied MB-based super-resolution FISH (MB-FISH) to visualize the endogenous target in mouse embryonic stem cell (ESC). Furthermore, we also applied CRISPR/Cas9-mediated knockout to generate a homozygous knockout (HoKO) negative control cell, in which a ~3 kb DNA fragment covering the 2.5- kb imaged target was deleted from both *Nanog* alleles. In each cell, we collected and analysed STORM imaging data from a layer (700 nm in depth) of the nucleus. Based on the previous version, we have now applied stringent strategies in obtaining nanostructures. For example, conventional imaging of beads is now provided to confirm that no identified nanostructure is overlapping with or close to fluorescent bead used for drift correction. After image reconstruction and candidate filtering, we detected 14 cells with MB-labelled nanostructure(s) from 92 WT ESCs but zero from 74 HoKO controls. When further quantifying the MB-FISH specificity by random sampling using the same criteria with larger samples, we found the p-value of 1.49e-04 for this endogenous experiment. In the future studies, more MB-labelled target sequence will be identified by collecting and analyzing complete datasets from the whole nucleus (at least 5 µm in depth) rather than only from a single nuclear layer (700 nm in depth) of each nucleus.

Second, although we had carried out STORM imaging in dozens of cells in the last submission, we have recently performed more in EGFP or blank SK-N-SH cells. During the revision, we identified 7 from 28 EGFP cells to be with detected nanostructure, one nanostructure in each positive cell; whereas zero from 33 blank controls. Collectively, we have so far identified 21 or zero cells to be with detected nanostructure from 78 EGFP or 83 blank cells, respectively, giving a p value of 1.00e-6 when random sampling using the same criteria with larger samples. Taken together, these data show the capability and robustness of our MB-FISH in visualizing a unique genomic sequence in the human or mouse genome. For the details, please see last two sections of Result and last three paragraphs of Discussion in the current version.

Finally, in regards to multiple lentiviral copies inserted at the same locus, a specific PCR band of 3.3 kb rather than 6.6 kb or larger in in Figure 1 had indicated no tandem copies at the same locus. In response to the reviewer’s concern, we designed a pair of outward-facing primers within the inserted region, with which no PCR band could be amplified if only single copy was inserted each locus (Figure 1—figure supplement 2). Specific band(s) was expected to appear from PCR using genomic DNA from EGFP rather than blank cells if two or more copies were inserted at the same site (Figure 1—figure supplement 2). It turned out that no specific PCR band were observed in amplification of EGFP-cell samples, indicating no tandem insertion in the same site (Figure 1—figure supplement 2). Related information could also be found in Results and Materials and method section (PCR confirmation of no tandem viral repeats integration) of the revised version.

Taken together, these data show the capability and robustness of our MB-FISH in visualizing a unique genomic sequence in the human or mouse genome.

*Major points:*

*1) Whether the non-complementary sequences could mimic the true off-targets in MB binding? It is difficult to design non-complementary sequences sharing 5-15 nucleotides with individual MBs to mimic many possible off-target MB bindings.*

When reading the fluorescence of a certain MB at the presence of its 42 nt complementary sequence (CS), we need a negative control, in which non-complementary sequences (NCSs) of the same length and concentration were added. In our experiment, the NCS for each MB was obtained by mixing CSs of other 28 MBs and shared 10 ± 5 nucleotide complementarity with this MB. Therefore, the results from these NCSs indicated that MB design decreased its fluorescing possibility to less than one quarter even at the presence of excessive amount of sequence sharing 10 ± 5 nucleotide complementarity with this probe, and we revised our description in the current Result section for (Figure 1).

*2) Whether the frame rate of 85 Hz and power of 0.5 mW 405-nm laser and 29 mW 641-nm laser are the best optimized condition since the values of blank cells are very low for calculating FDR? In addition, in Figure 2, 0.04 ± 0.02 basically means there is no auto-fluorescence.*

In fact, auto-fluorescence in blank/control samples at 20 Hz frame rate cannot be neglected, almost equal to 34-55% of the events identified from Alexa-647-labeled samples under the same condition (see Condition I in revised Figure 2). Hence, we titrated multiple imaging conditions (frame rate and laser powers) to find out an optimal condition, under which auto-fluorescence was mostly minimized while the blinking property of Alexa-647 was maintained at a normal state. As we can see from white bar of conditions IX to XII in revised Figure 2 (previous Figure 2), autofluorescence was stably minimized to a very low level that was mentioned by the reviewer as “no fluorescence”, only when the imaging frame rate was increased to 85 Hz. Thus, the lowest FDR from condition XII indicated it as the optimal one. Even under an unideal buffer, condition XII was really to have a lowest FDR, confirming the robustness of this condition as the most optimal (see revised Figure 2). Furthermore, we also compared Condition XII to others via evaluating the photo-bleaching effects and measuring Alexa-647 detection efficiency, from which Condition XII is consistently shown to be optimal. Moreover, this optimized condition has been applied to super-resolution imaging of a 2.5-kb integrated or endogenous target in the nuclear genome, leading to identification of fine-scale nanosctructures in the positive cells rather than the negative controls. Collectively, the frame rate of 85 Hz and power of 0.5 mW 405-nm laser and 29 mW 641-nm laser are regarded as the best optimized conditionAlexa-647 imaging in cell autofluorescence background. For your information, the laser powers in each condition are now present in format of power density in the revised version according to another reviewer’s suggestion.

*3) It would be useful if the authors could show that the method could be used to study an enhancer-promoter or promoter-promoter looping contact as a proof-of-principle.*

In response to the reviewer's comment, we carried out superresolution imaging of a secondary target besides the integrated DNA in human genome. This is a 2.5 kb endogenous genomic element and a non-coding region that has been disclosed as an ESC-specific super-enhancer within mouse Nanog locus (Blinka et al., 2016). Combining STORM imaging to MB-FISH with a set of 34 MBs designed according to the target sequence, we successfully visualized this super-enhancer in mouse embryonic stem cell (ESC). The nanostructures observed were specific since they were not observed in the homozygous knockout (HoKO) ESC, in which a ~3 kb DNA fragment covering the 2.5- kb imaged target was deleted from both Nanog alleles. This work provided a basic for further visualization of looping between this super-enhancer and the Nanog promoter in mouse ESC.

*Reviewer #4:*

*In their manuscript, Zhang et al. develop a FISH approach based on molecular beacons that can be used to label short (2.5kb) non-repetitive regions of genomic loci for 3D super-resolution imaging. The approach is interesting and can provide a simpler method of labeling non-repetitive sequences compared to Oligopaint approach. However, I have major concerns about the specificity of the probes and the validity of the method (see comments below):*

*Major comments:*

*1) The major weakness of the manuscript is the use of cells with random integration instead of a more controlled system. This choice makes it difficult to truly assess the specificity of the approach. The authors should have used a system of site specific integration (for example: CRISPR-Cas9, FlpIn recombination, Integrations in R26 locus). Alternatively, the authors could have targeted an endogenous region and use as control the knockout counterpart. Wild type vs KO cell lines are available for a variety of genes or can be generated with CRISPR-Cas9. In these systems the authors should detect a number of loci compatible with the ploidy of the cell. In addition, in such a system the authors can confirm the specificity of the molecular beacons by labeling the same genomic locus with an alternative method such as regular FISH. In the absence of these experiments, the specificity of the molecular beacons remains unconfirmed and not convincing from the provided data.*

We appreciate the reviewer for the comments and suggestion, in response to which we have performed substantial experiments and remarkably improved our work.

First, we chose a 2.5-kb non-coding genomic region (a super-enhancer within *Nanog* locus) (Blinka et al., 2016) in the mouse genome as our new target, designed a set of 34 MBs according to the target sequence, and applied MB-based super-resolution FISH (MB-FISH) to visualize the endogenous target in mouse embryonic stem cell (ESC). Following the reviewer’s suggestion, we applied CRISPR/Cas9-mediated knockout to generate a homozygous knockout (HoKO) negative control cell, in which a ~3 kb DNA fragment covering the 2.5- kb imaged target was deleted from both *Nanog* alleles. In each cell, we collected and analysed STORM imaging data from a layer (700 nm in depth) of the nucleus. Based on the previous version, we have now applied stringent strategies in obtaining nanostructures. For example, conventional imaging of beads is now provided to confirm that no identified nanostructure is overlapping with or close to fluorescent bead used for drift correction. After image reconstruction and candidate filtering, we detected 14 cells with MB-labelled nanostructure(s) from 92 WT ESCs but zero from 74 HoKO controls. When further quantifying the MB-FISH specificity by random sampling using the same criteria with larger samples, we found the p-value of 1.49e-04 for this endogenous experiment. In the future studies, more MB-labelled target sequence will be identified by collecting and analyzing complete datasets from the whole nucleus (at least 5 µm in depth) rather than only from a single nuclear layer (700 nm in depth) of each nucleus.

Collectively, these data show the capability and robustness of our MB-FISH in visualizing a unique genomic sequence in the human or mouse genome.

*2) The manuscript is also largely descriptive and lacking quantitative information to back up the major claims. For example, was the viral MOI (multiplicity of infection) assessed? From this value at least a range of expected integration sites could be estimated and compared to the super-resolution data.*

*The authors should also provide more information regarding the efficiency of their approach:*

*– what is the percentage of GFP positive cells with detectable loci?*

*– what is the average number of detected loci? Is this number in line with the viral MOI used?*

*– what is the average and range of detected localisations per locus?*

*In general, representative images should be supported with more quantitative data based on the complete dataset of cells imaged. It would also be useful to provide unzoomed images to appreciate the general signal to noise ratio in the whole ROI.*

Thanks the reviewer for the comments and detailed suggestions. As a response, we included more quantitative information and provided representative images in the revised version. In regards to “the percentage of GFP positive cells with detectable loci”, it was 92.4 ± 1.5% as described in the revised manuscript. Furthermore, we showed in the current version that 21 from 78 EGFP cells were identified with detected nanostructure(s) but zero was recognized from 83 blank cells. When further quantifying the MB-FISH specificity by random sampling using the same criteria with larger samples, we found the p-value of 1.00e-6 in visualizing this 2.5 kb integrated DNA in human genome. The number of detected loci in each individual cell is still not clear, since we failed in obtaining EGFP cells of single colonies and the integrated locus number varied from cell to cell. Moreover, we showed in the current version that 14 from 92 WT ESCs cells were identified with MB-labelled nanostructures but zero from 74 HoKO controls, giving a p value of 1.49e-04 when random sampling using the same criteria with larger samples. Related details are included into the revised Result section. Finally, unzoomed images from bright field or under 405-nm or weak 641-nm laser illuminations were provided to show the general signal to noise ratio in the cell nucleus as well as the position of fluorescence beads (Rows A-C, Figure 4 and Figure 5). Taken together, these quantitative data substantially support the capability and specificity of this MB-FISH in visualizing short unique genomic elements.

*3) When comparing MB-FISH to Oligopaints, the authors should further discuss the differences/similarities between the two approaches:*

*– can MB-FISH provide allele specificity and is it sensitive to SNPs.?*

*– what is the optimal probe density required for MB-FISH?*

*– are there specific requirements in terms of spacing between probes, strand specific orientation, etc?*

*– would this approach be compatible with live imaging?*

Thank the reviewer for the constructive suggestions. In response to these discussions, we have an intense discussion about the similarities and differences between Oligopaint FISH and MB-FISH, which is now included in the first and second paragraphs of revised Discussion.

*4) The usage of English language in the manuscript needs substantial improvement. The grammatical errors and awkward sentences are far too many to point out one by one in this review.*

We thank the reviewer for the patience and suggestion. We have seriously revised manuscript and hoped that the current version could be better understood.

[Editors’ note: the author responses to the re-review follow.]

*Reviewer #1:*

*The authors addressed most issues raised in my first review. However to meet the highest standards of eLife, a few minor issues still need to be resolved:*

*1) Sections of the manuscript have become somewhat cluttered and difficult to digest, in part by addressing the reviewers' additional requests. For instance, experimental details in the Results section (e.g. cloning controls) could potentially be moved to the Methods section. Also the Discussion could be streamlined (see also point 3). Generally I feel the manuscript would benefit from another round of decluttering/shortening to present this excellent work in the most concise and clearest way.*

Thank the reviewer for the suggestion. As a response, we move the description about tandem integration exclusion to the Materials and method section (PCR exclusion of tandem viral repeat integration) of the current version. We completely agree with the reviewer on the emphasis of a most concise and clearest expression. But, it is hard to remarkably shorten this section while remaining the complete responses in the revised manuscript, since a lot of points have been added to the Discussion as responses reviewers' comments. Nevertheless, we have thoroughly edited the draft and streamlined part of the Discussion, including the fourth to sixth paragraphs, to facilitate its digestion by the audience.

*2) I am still not entirely clear how to interpret the success rate of 21/78 and 14/92 cells with detected nanostructures. Does this reflect the limitation of the STORM imaging method to be able of imaging only a sub-volume of a mammalian nucleus? Or does is reflect (also) a lowered detection efficiency of the FISH method. Or is it a combination of both? What is the chance of detecting positive cells, if one assumes one, two (or more) loci, if the detection would be 100% efficient? The authors may still need to clarify this better.*

Using this MB-FISH method, we observed 21 from 78 EGFP cells and 14 from 92 ES cells with detectable nanostructures. In addition to MB labeling efficiency, this detection rate is also contributed by the following two aspects. On the one hand, in this proof-of-principle study, we only collected images from a single nuclear layer (700 nm in depth) per cell. This does NOT reflect the limitation of the STORM imaging method to be able of imaging only a sub-volume of a mammalian nucleus (collecting all layers of the whole nucleus is a regular rather than challenging task for STORM imaging). This is only due to the limited time in the manuscript revision, during which we had selected a second endogenous target, designed/synthesized the series of MB probes for the new target, generated a HoKO cell line, found out the experimental conditions for preparing and imaging the round and weakly adherent ESC samples, and carried out the MB labeling and STORM imaging experiments. Eventually, this substantial work had extended our re-submission deadline from two months to 100 days. In practice, higher percentage of cells with positive nanostructure(s) could be observed by collecting multiple layers (e.g. 5 layers) from each nucleus (at least 5 µm in depth). On the other hand, cells undergoing mitosis are insensitive to either Oligopaint or MB labeling, since nuclear genome in these cells is in form of chromosome, a highly compacted structure with genome topology different from that in interphase.

In response to the reviewer’s concern, we revised paragraph four of current Discussion section.

*3) The use of language needs improvement (particular in the revised discussion).*

We thank the reviewer for his patience and detailed suggestions. As a response, we have revised the text and corrected some mistakes.

*Reviewer #2:*

*The majority of my questions have been answered, in particular the authors have analysed an endogenous locus. However there remain outstanding issues that require explanation and/or addressing in the text for me to feel confident of the validity of the signals presented. Criteria for image analysis and some of the data produced do not appear to be presented fully.*

*1) Response to Reviewer 2 Point 6: In fact the Markaki 2012 paper clearly states (page 415, last paragraph): "In agreement with our visual impression, the IC space was reduced in 3D-FISH-treated nuclei and a shift toward higher intensity classes. (…) We also consider the possibility that some swelling or dispersal of chromatin, in particular as a result of the heat denaturation step [18], resulted in an improved accessibility of DNA to this fluorophore." Please could the authors clarify this point in the Discussion.*

This study establishes a method to detect non-repetitive short genomic elements at nanoscale resolution in single cells, based on the widely-used oligonucleotide-based FISH. Its key principle is to enable probe accessibility to the target sequence, while maintaining the chromatin architecture via forming formaldehyde-mediated covalent cross-links between spatially close DNA and/or proteins. With a long-term goal of visualizing topologic structure (connector) rather than geometrical structure (size, shape), this study puts great efforts to avoid reverse dissociation of formaldehyde cross-links in the post-fixation procedures. Given this specific aim, the chromatin topology could be maintained relatively well if the covalent cross-links in the fixed cell samples are not disrupted, even though there is a minimal level of inter-chromatin space reduction or chromatin swelling induced by protected freeze-thawing as well as heat denaturation (Markaki et al., 2012, Solovei et al., 2002).

The formaldehyde-mediated cross-link is sensitive to heat rather than low temperature (liquid nitrogen) or organic solvent (formamide or ethanol). To minimize the cross-link reversals in fixed samples and hence to maintain the nuclear ultrastructure, we perform hybridization at remarkably low temperature (22°C), compared to 42°C -47°C in Oligopaint-FISH (Beliveau et al., 2015, Beliveau et al., 2012, Boettiger et al., 2016, Wang et al., 2016) or 37°C in the paper mentioned by the reviewer (Markaki et al., 2012). As indicated in Figure 2 from a previous report by Kennedy-Darling and Smith (2014), significantly less DNA was dissociated during 20-h hybridization at 23°C than that at 37°C or at 47°C, indicating less disruption of formaldehyde cross-links under our hybridization condition. In addition, we also decrease temperature of FISH washing step and remove the 0.1N HCl treatment as well for our MB-FISH. With these modified conditions, chromatin topology in MB-FISH is expected to be maintained relatively well, compared to other methods like Oligopaint-FISH that was reported to provide results consistent with chromosome-conformation capture technology data (Wang et al., 2016). Unless the cell is frozen (e.g. cryo-EM), it is so far hard to find a better way to preserve exact 3D chromatin structure (Branco et al., 2008). It would be great if we could combine MB-labeling with cryo-EM technique in the future.

In response to the reviewer’s suggestion, we added the explanation of this point as the last Discussion paragraph of the current manuscript.

*2) Results and Figure 4 wondered why the beads (F8810 580/605) were visible at 405nm and occasionally at 641nm?*

Thank the reviewer for the question. According to the product information of the beads (F8810, Thermo Fisher), their excitation efficiency at 641-nm laser is much lower than that at 405-nm laser (efficiency at 641-nm vs 405-nm: 0.14 vs 1) and is relatively unstable. In addition, although the beads were supposed to be generally uniform as claimed by manufacturer, some of their properties are unavoidably heterogeneous at some levels. This is reflected by varied brightness among different beads of the same batch even in 405-nm laser imaging. As a result, the beads could be stably visualized at 405-nm laser, but occasionally detected at 641-nm laser due to the low excitation efficiency and heterogeneous properties.

In response to the reviewer's comment, we have revised the description of beads to be “the fluorescent beads visible stably at 405 nm and occasionally at 641 nm due to weak excitation and heterogeneous bead properties” in the Result section paragraph eleven.

*3) Results subsection “Super-resolution visualization of 2.5 kb enhancer in situ in Nanog locus of mouse ESCs” and Figure 5—figure supplement 1: It is not clear what point the authors are making about the nuclear periphery sitting close to the cell surface. Please clarify.*

To determine if a potential nanostructure localizes within nucleus, we collect bright-field image of SK-N-SH cell to determine the nuclear periphery. The SK-N-SH cell is large and flattened, and its nuclear membrane outline could be readily recognized under bright field. But ESCs are round and partly-adherent, and their nuclear peripheries are hardly recognized via bright field imaging. In this regard, we performed conventional co-imaging of bright field and DAPI (Figure 5—figure supplement 1) before STORM imaging to delineate/determine the nuclear periphery of each cell as stated in the text and Figure 5—figure supplement 1. We found the nuclear edge was very close to cell outline that was visible under bright field. Therefore, we highlight the cellular outline of ESCs to approximately indicate the nuclear edge, so that we could determine if a nanostructure localizes within nucleus in Figure 5.

In response to reviewer’s comment, we revised the description to better explain our aim at pointing out the observations about ESC’s nuclear periphery sitting close the cell surface.

*4) Figure 4: What is the blue dot in Cell I E?*

The blue dot in Cell I E (Figure 4) is a single switching event from a depth of -159.58 nm in z dimension, which is corresponding to pseudo-colour of blue. This event is at long distance (around 270 nm) from the identified nanostructure (yellow) in the same view of field and thus is excluded as a wild event, although it is still plotted in the reconstructed image to avoid erasing any detected events with bias.

*5) Figure 5 Panel C Cell V: The position of the nanostructure in the box in C at low resolution does not appear to be the same as in D. Please explain.*

We thank the reviewer for his/her careful reading and kind reminder. In previous Figure 5 Cell V, Panel D did not exactly match the region of green box in Panel C (180 nm away from each other). In the revised Figure, we have corrected this mistake by plotting the nanostructure from the right corresponding region.

*6) Results: The authors provide statistical data on larger series than the 14/92 ESCs and the 21/78EGFP cells. Why are the numbers in the larger series not provided?*

Thank the reviewer for the reminder. To assess the specificity of MB-FISH in positive and negative control samples, we had calculated p value using random sampling in 1×10^[6]^ sample size in the previous submission, leading to a p value of 1.49×10^-4^ for ESCs and 1.0×10^-6^ for EGFP cells. In fact, when the sample size was increased to 1×10^[8]^, the p values have been improved to be more accurate (14/92 for ESCs: p=1.5×10^-4^, 21/78 for EGFP cells: p=5.4×10^-8^). Furthermore, we measured the p value based on the cumulative distribution function (CDF) of the hypergeometric distribution, similarly leading to a p value of 1.6×10^-4^ for ESCs and 5.0×10^-8^ for EGFP cells. All the p values calculated indicated that the specificity of the MB-FISH was significant. In the current version, we present the hypergeometric distribution p values which are consistent with the p values from random sampling in 1×10^[8]^ sample size.

*7) Discussion paragraph three: Please give the limits in localisation number and area that were used as criteria for defining nanostructures.*

Thank the reviewer for the reminder. The area upper limit was empirically determined to be 0.2 μm^[2]^ in our nanostructure identification criteria. The upper limit of localization number was 3000, which was set based on the following two issues. 1) The maximal probe number on each target sequence was around 30 (up to 34 or 29 probes for endogenous or inserted viral sequence, respectively); and 2) The statistical data of 100-150 single Alexa-647 molecules revealed that each Alexa-647 was detected on average for a few tens of times under our current imaging condition, with a maximal number of 100 detection times (data not shown). Thanks to the reviewer's reminder, the values of limits in localization number and area are now added into the revised Discussion section (paragraph three).

*8) Discussion paragraph five: This explanation could perhaps be clarified.*

We thank the reviewer very much for the comment. During addressing this comment, we have revised and streamlined paragraph five of the Discussion section to make it more concise and easier for digestion, as it was also strongly suggested by another reviewer.

This paragraph mainly focuses on discussing the various observation of the same nuclear region subjected to conventional or STORM imaging (Row C and D in Figure 4 and Figure 5), in which a 641-nm laser is used to activate MB fluorophores. In most cases, STORM nanostructures with large or small amount of localizations appear in 641-nm conventional imaging as fluorescent dots (Cell I, III in Figure 4 and Cell I, II in Figure 5) or unrecognizable structures/non-obvious clusters (Cell II in Figure 4 and Cell III, IV, VI in Figure 5), respectively. These are generally consistent with common sense. But in a small number of other cases, those fluorescent dots visible in 641-nm conventional imaging turn out in STORM imaging to be nanostructures without lots of localizations (Cell V in Figure 5) or to be excluded from the nanostructure candidates due to a sparse distribution or an extremely low amount of localizations (See Figure 5—figure supplement 4). These are somehow unexpected and look "inconsistent", since fluorescent dot visible in conventional imaging is supposed to contain at least a certain number of fluorophores and expected to exhibit relatively high localization number in STORM imaging. These "inconsistent" observations are possibly due to the relatively lower blinking frequency of fluorophores on that nanostructure.

Thanks to the reviewer's comments, we have substantially revised paragraph five of the Discussion section to clearly deliver these points. In addition, we corrected the mistake in Figure 5, in which two conventional images (panel B and C) of cell III illuminated under 405- nm or 641- nm laser were mis-interchanged in the previous version.

*9) Discussion paragraph six: The authors could perhaps refer to Ricci et al. here (PMID: 25768910).*

We thank the reviewer for the suggestion. As a response, we included the work of Ricci et al., 2015 in paragraph six of the Discussion section.